# Continuous millisecond conformational cycle of a DEAH box helicase reveals control of domain motions by atomic-scale transitions

Robert A. Becker[1] & Jochen S. Hub [1✉]

Helicases are motor enzymes found in every living organism and viruses, where they maintain the stability of the genome and control against false recombination. The DEAH-box helicase Prp43 plays a crucial role in pre-mRNA splicing in unicellular organisms by translocating single-stranded RNA. The molecular mechanisms and conformational transitions of helicases are not understood at the atomic level. We present a complete conformational cycle of RNA translocation by Prp43 in atomic detail based on molecular dynamics simulations. To enable the sampling of such complex transition on the millisecond timescale, we combined two enhanced sampling techniques, namely simulated tempering and adaptive sampling guided by crystallographic data. During RNA translocation, the center-of-mass motions of the RecA-like domains followed the established inchworm model, whereas the domains crawled along the RNA in a caterpillar-like movement, suggesting an inchworm/caterpillar model. However, this crawling required a complex sequence of atomic-scale transitions involving the release of an arginine finger from the ATP pocket, stepping of the hook-loop and hook-turn motifs along the RNA backbone, and several others. These findings highlight that large-scale domain dynamics may be controlled by complex sequences of atomic-scale transitions.

[1] Theoretical Physics and Center for Biophysics, Saarland University, Saarbrücken, Germany. ✉email: jochen.hub@uni-saarland.de

Since the discovery of helicases in *E. Coli* in the 1970s, helicases were found in all organisms including viruses[1,2]. They play crucial roles in processes involving DNA or RNA such as DNA transcription, translation, recombination, repair, proofreading, ribosome biogenesis, RNA transport, splicing, degradation, assembly, and disassembly. Helicases use the energy from nucleoside triphosphate (NTP) hydrolysis either to unwind double-stranded RNA/DNA or to drive the translocation of RNA/DNA, as studied in detail in recent decades[3–19]. Due to their critical role in genome maintenance, helicases are involved in genetic diseases such as cancer or in aging-related disorders such as Bloom syndrome[20,21], Werner syndrome[22], and many others[23–27].

Helicases have been grouped into six super-families and various subfamilies based on conserved structural motifs or domain similarities[4]. The super-family 2 (SF2) forms the largest super-family and includes, among others, the DExD/H-box, RecQ-like, and Snf helicases[28–31]. These have further been classified into up-and downstream helicases, denoted as type A and B, respectively, and into helicases working on single- or double-stranded ligands, denoted as α and β types, respectively[4]. Like in many other NTP-binding proteins, the Walker A/B motifs, also known as P-loop, is highly conserved among helicases[32]. In DexD/H helicases, more specifically, the ATP binding pocket is defined by two recombination protein recombinase A-like (RecA-like) domains that must arrange in close proximity to carry out ATP hydrolysis (Fig. 1)[33,34].

The main function of conventional helicases is the unwinding of double-stranded DNA mediated by ATP hydrolysis, however certain helicases instead translocate single-stranded DNA or RNA (ssRNA), establishing them rather as translocases[4]. Two such translocating SF2Aα DEAH-box helicases are Prp22 and Prp43, which move directionally along the RNA strand by binding to the RNA phosphodiester backbone. X-ray crystallographic studies revealed the structures of Prp22 and Prp43 from *C. thermophilum*, providing insight into the ATP binding and hydrolysis

mechanism as well as into certain snapshots along the RNA translocation cycle[35,36]. However, because many intermediate conformations along the cycle are unknown and because of a lack of RNA-bound crystal structures, the understanding of the relevant domain motions and the cascades of the molecular switches remains limited.

The widely discussed inchworm model describes RNA translocation by helicases with a stepping process[4,28,37]. Accordingly, while the first RecA-like domain is tightly bound to the RNA, the second RecA-like domain may move along the RNA driven by a power stroke from NTP hydrolysis until it finds a new tightly binding contact. Next, by another change of the NTP-binding state, the first RecA-like domain is only weakly bound, enabling it to follow the previously moved RecA-like domain along the RNA. Thus, according to the inchworm model, at least one RecA-like domain is tightly bound to the RNA at any time while the other RecA-like domain changes its affinity to the RNA depending on the NTP-binding state.

Molecular dynamics (MD) simulations have been used to follow conformational transitions of enzymes. However, equilibrium MD simulations of complex conformational cycles involving multiple molecular switches are challenging because the simulations typically do not cover the functionally relevant timescales. Pulling simulations may in principle overcome long timescales; however, they are often not applicable because they require the definition of a good reaction coordinate for the process, which is far from trivial for complex, multi-step, non-linear conformational transitions. In this computational study, we obtained a complete conformational cycle of RNA translocation in Prp43 by combining two enhanced sampling techniques, namely simulated tempering (ST) and adaptive sampling (AS). ST may accelerate the sampling of transitions by approximately one order of magnitude while maintaining the correct Boltzmann distribution and without the need of defining a reaction coordinate[38,39]. Instead, ST enhances the sampling by switching the temperature of the system along a pre-defined temperature ladder with a Metropolis criterion. Because enthalpic barriers are flattened at higher temperature, the system may carry out transitions in shorter simulation times as compared to a standard simulation at room temperature.

For AS, multiple rounds of short parallel simulations are carried out[40]. After each round, the most "successful" simulations are selected as seed for the next round based on their progress along a set of pre-selected structural features such as the presence of H-bonds, distances, or angles to reach a pre-defined reference conformation. The main advantage of the AS technique is the ability to parallelize simulations and to guide the sampling towards undersampled regions of conformational space, which both can lead to a drastic decrease in the overall wall time. In addition, AS may enhance the sampling if the rate-limiting transitions are slow and spatially clustered[41]. AS is similar to the supervised molecular dynamics (SuMD) approach, which has been used to enhance the sampling of ligand binding simulations[42,43]. In contrast to frequently used autonomous supervising algorithms, we have chosen the "successful" AS simulations by human supervision by carefully investigating the features after each round of the AS (Supplementary Fig. S1). Simplified flow charts of the AS and ST are shown in Supplementary Figs. S15 and S16.

## Results
### Simulating a complete Prp43 conformational cycle by adaptive sampling.
RNA translocation in Prp43 involves large domain motions, which are coupled to small side-chain rearrangements at the protein–RNA and the protein–protein interfaces. Even

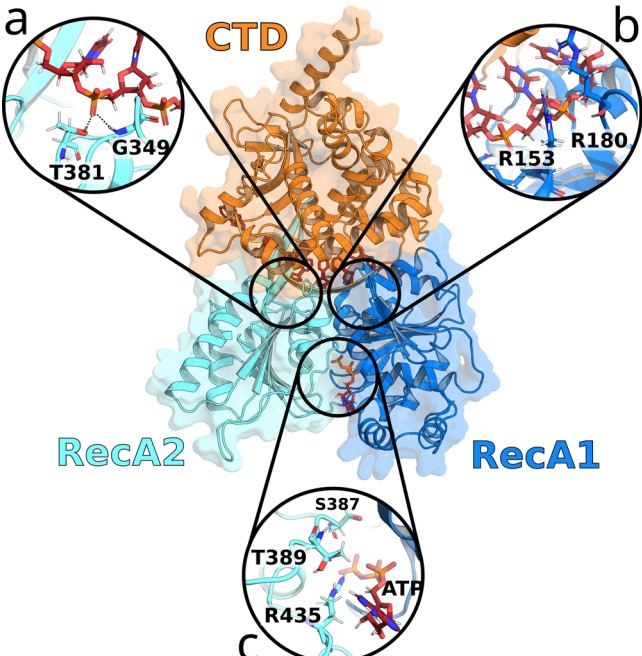

**Fig. 1 Prp43 in the closed state (PDB ID: 5LTA).** C-terminal domain (CTD, orange), RecA2 domain (cyan) and RecA1 domain (blue). Boxes show close-up views on the hook-loop (**a**), the hook-turn (**b**), and on the ATP-pocket (**c**).

**Table 1 Structural features used to monitor the progress of opening and closing transitions.**

| Feature | [unit] | Opening | | | Closing | | |
|---|---|---|---|---|---|---|---|
| | | 5LTA | 6I3P | FoO | FoO | s5LTA* | FoC |
| RecA1 COM – COM RecA2 | [nm] | 2.65 | 3.15 | 3.12 | 3.12 | 2.65 | 2.67 |
| G349 H – $O^{1A}$ U5 | [nm] | 0.19 | 0.71 | 0.70 | 0.70 | 0.70 | 0.74 |
| T381 $H^\gamma$ – $O^{1A}$ U5 | [nm] | 0.26 | 0.75 | 0.87 | 0.87 | 0.81 | 1.03 |
| R435 $H^\eta$ – ATP | [nm] | – | – | – | 0.75 | 0.38 | 0.40 |
| K403 $H^\zeta$ – $O^{1A}$ U4 | [nm] | 0.29 | 0.96 | 0.77 | 0.77 | 0.91 | 0.93 |
| E316 H – $O^{1A}$ U4 | [nm] | 0.21 | 0.72 | 0.71 | 0.71 | 0.70 | 0.73 |
| S387 $H^\gamma$ – Mg | [nm] | – | – | – | 1.44 | 0.26 | 0.49 |
| S387 $H^\gamma$ – $O^{1A}$ U5 | [nm] | 1.58 | 0.19 | 0.17 | 0.17 | 1.42 | 1.18 |
| G349 H – $O^{1A}$ U4 | [nm] | 0.92 | 0.24 | 0.21 | 0.21 | 0.17 | 0.20 |
| G349 H – $O^{2A}$ U4 | [nm] | 0.73 | 0.31 | 0.30 | 0.30 | 0.32 | 0.32 |
| T381 $H^\gamma$ – $O^{1A}$ U4 | [nm] | 0.90 | 0.17 | 0.17 | 0.17 | 0.16 | 0.18 |
| T381 $H^\gamma$ – $O^{2A}$ U4 | [nm] | 0.73 | 0.38 | 0.42 | 0.42 | 0.40 | 0.42 |
| R180 H – $O^{1A}$ U5 | [nm] | – | – | – | 0.83 | 0.18 | 0.20 |
| R153 H – $O^{1A}$ U5 | [nm] | – | – | – | 0.62 | 0.19 | 0.18 |
| R180 C – $O^{1A}$ U7 | [nm] | – | – | – | 0.48 | 0.90 | 1.04 |
| K403 $H^\zeta$ – $O^{1A}$ U3 | [nm] | 0.87 | 0.27 | 0.28 | 0.28 | 0.33 | 0.28 |
| S387 Phi | [°] | 59 | −65 | −68 | −68 | 59 | 51 |
| S387 Psi | [°] | −47 | −47 | −33 | −33 | −47 | 43 |

Opening: Starting values taken from the closed crystal structure of Prp43 (PDB ID 5LTA), target values taken from open structure of Prp22 (PDB ID 6I3P), and final values of simulated opening process (FoO). Closing: Starting values from the final opening simulation frame (FoO), target values from the closed crystal structure of Prp43 with the RNA shifted by one nucleotide, and final values of the simulated closing process (FoC). Overall, the features of the final simulation frames of the opening and closing are in good agreement with the reference values for the corresponding features.

individual rearrangements of the multi-step process may occur on the microsecond timescale, rendering simulations of the complete cycle intractable with conventional MD simulations. Hence, we used AS to simulate RNA translocation in Prp43, while we enhanced the sampling of individual AS simulations with ST.

We decomposed the translocation cycle into two major transitions: (i) the "opening transition" involving the opening of the RecA1–RecA2 interface and the sliding of RecA2 along the RNA by one base pair; and (ii) the "closing transition", characterized by the closure of the RecA1–RecA2 interface and the sliding of the RNA along RecA1. AS of the opening process started from the Prp43•$U_7$•ATP structure (PDB code 5LTA[35], Fig. 1) and was triggered by the removal of ATP, thereby modeling Prp43•$U_7$ after the dissociation of the hydrolyzed ATP. Since the presence of ATP stabilized cationic moieties at the RecA1/RecA2 interface, removal of ATP led to a electrostatic repulsion between the two RecA-like domains. To monitor the opening transition, we selected a set of 18 structural features whose target values were taken from an open structure of the homologous Prp22 (Table 1, PDB ID 6I3P[36]). The opening process was completed after nine rounds of AS, as evident from a reasonable agreement of the structural features with their target values (Table 1, middle columns). Here, each round was composed of 40 to 500 simulated tempering simulations of 10 to 100 ns. The concatenated successful simulations summed up to a simulation time of 580 ns, whereas the invested overall simulation time was 56.5 $\mu$s.

The closing simulation started from the final frame of the opening simulation and was triggered by inserting ATP into the binding pocket of the RecA1 domain. We monitored the progression of the closing transitions by comparing the selected structural features with their values in the closed Prp43•$U_7$•ATP complex (PDB code 5LTA). The closing transition was completed after 13 rounds of AS, again revealed by a reasonable agreement of the structural features with their respective target values (Table 1, last columns). A typical table used to monitor the progression of the structural features is shown in Supplementary Fig. S1. Here, each round involved 10 to 50 individual simulations of 10 to 100 ns each, summing up to an total simulation time of 40 $\mu$s, while the concatenated successful

simulations summed up to 1.2 $\mu$s. The increased simulation time of the concatenated trajectory as compared to the opening simulation likely reflects that the formation of a well-defined protein–protein interface is more challenging than the rupture of an interface.

In summary, by using AS augmented with ST, we obtained a complete conformational cycle of the motor enzyme Prp43. As described in the following, the concatenated successful simulations provided an unprecedented atomic view on the function of a helicase, involving large-scale domain motions as well as the atomic-level rearrangements that were required for RNA translocation.

**Large-scale domain motions.** Figure 2a–c shows the motions of the RecA2 and the C-terminal domain relative to the RecA1 domain during the opening transition. After release of the ATP, the center-of-mass (COM) distance between RecA1 and RecA2 increased rapidly by 0.3 nm within the first 80 ns of the cumulative simulation time of successful AS simulations. Here, the cumulative simulation time of successful AS runs should not be confused with the by far longer physical time that would be required to observe such transition by a single conventional MD simulation (Fig. 2c). The RecA1–RecA2 COM distance exhibited a second sudden increase at 200 ns, followed by a gradual relaxation towards the value of the open conformation of the homologous Prp22 structure. The ongoing fluctuations of the open state are in line with the crystallographic data by Ficner et al.[36], who observed high B-factors of the RecA2 domains in the open Prp22 structure. The sudden changes of the RecA1–RecA2 distance correlate with transitions of molecular switches discussed below, suggesting that large-scale domain motions are controlled by atomic-scale transitions of molecular switches.

Previous crystallographic data revealed that the C-terminal domain (CTD) may carry out large-scale motions relative to RecA1 and RecA2, which are likely required for loading of the RNA into the RNA tunnel along the interface between CTD and RecA1/RecA2 (proposed by Taubert et al.[35], see also Fig. 2a, d). By visual inspection of the simulations and by analyzing the center-of-mass motions of the CTD relative to RecA1 and RecA2, we found that CTD and RecA2 translate concertedly along the RNA during the

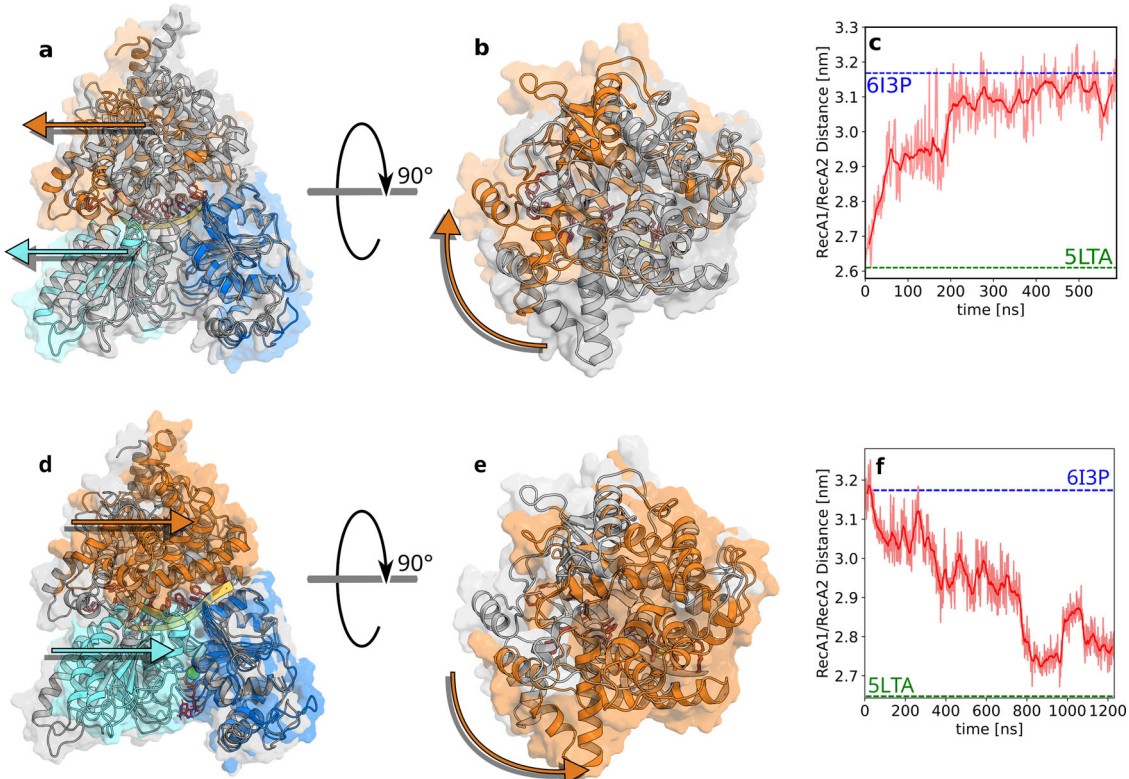

**Fig. 2 Domain movements during the conformational cycle. a–c** Opening transition and (**d–f**) closing transition of Prp43. **a**, **d** Front view and (**b**, **e**) top view at the beginning (gray) and end (multi-colored) of the respective process taken from the initial and final frames of the opening or closing trajectory, respectively. Arrows highlight the motions of RecA2 domain (cyan) and CTD domain (orange) relative to RecA1 domain (blue). **c** Center-of-mass distance between RecA1 and RecA2 domains during opening and (**f**) during closing. Dashed lines indicate the RecA1–RecA2 distances in the closed Prp43 structure (green, pdb code 5LTA) or in the open structure of the homologous Prp22 (blue, pdb code 6I3P). The time averages are drawn as dark red lines while the underlying data is drawn in pale red.

opening transition while RecA1 remains bound to the RNA (Fig. 2a, arrows; Supplementary Fig. S2). In addition to the center-of-mass displacement of the CTD relative to RecA1, the CTD carried out a rotation around a hinge located at the backside of the enzyme (Fig. 2b, arrow). This rotation is compatible with the presence of different CTD arrangements observed by X-ray crystallography in different ligand states of Prp43[35].

During the closing process, the overall domain motions were reversed relative to the opening transition, characterized by a concerted motion of CTD and RecA2 relative to RecA1 (Supplementary Fig. S2). However, in contrast to the opening process, the RNA was tightly bound to RecA2 in the closing process, thereby translocating by one nucleotide along RecA1. The RecA1–RecA2 distance decreased in three major steps at 300 ns, 750 ns, and at 880 ns (Fig. 2f), which again correlated with transitions of molecular switches discussed below. The final RecA1–RecA2 distance was in excellent agreement with the Prp43•U₇•ATP structure, suggesting that the RNA translocation cycle was completed. The resulting successful trajectory of all the transitions can be seen in the Supplementary Movie S2, which resembles the concatenated trajectories of the opening and closing processes. In addition, we plotted the RMSD of the opening and closing trajectory relative to the crystal structure 5LTA by fitting the structures onto the RecA1 domain (Supplementary Fig. S8). Here, we see a similar trend like the RecA distances shown in Fig. 2.

**Kinetic models of opening and closing processes.** We used two complementary models to obtain the approximate kinetics of the

opening and closing processes. First, based on the formalism by Pande and Singhal[44], we modeled the Prp43 dynamics by a linear sequence of transitions, providing an intuitive, simple, and numerically robust kinetic model of the opening and closing processes. Second, we constructed a Markov state model (MSM) which provides, in addition to the approximate kinetics, a view on the underlying free energy landscape and on the conformations of metastable states[45–49]. While MSMs rely on an elaborate theory for dimensionality reduction and kinetic modeling, MSMs have been shown to be sensitive with respect to limited sampling, as common when simulating complex protein dynamics[50].

Based on the formalism by Pande and Singhal[44], we modeled both the opening and closing process as seven-step processes, and we estimated the transition state matrix from the successful AS simulations (see Methods and Materials). For the opening and closing processes, we obtained MFPTs in the order of ~100 μs and ~50 μs, respectively. Assuming that ST accelerates the kinetics by one order of magnitude[39], these values translate into physical MFPTs in the order of 1 ms and 0.5 ms, respectively. This implies a maximum translocation speed of ~600 base pairs per second (bp/s). Owing to contributions from ATP binding, hydrolysis, and release, which may even be rate-limiting for the overall cycle, the translocation speed of Prp43 is likely lower than the maximum value of ~600 bp/s estimated from our simulations. Notably, assuming some reduction of the translocation speed from ATP binding, hydrolysis and release, our value is in reasonable agreement with translocation speeds of 100 to 300 bp/s observed for other helicases[51–53].

Complementary, we derived an MSM of the overall conformational cycle by combining all AS trajectories. The free energy

landscape projected onto two independent components revealed several well-separated metastable states, as visualized in Supplementary Fig. S9. The MSM suggested a MFPT of the closing process of ~ 40 μs, in reasonable agreement with the value of 50 μs obtained by the linear kinetic model. Furthermore, our MSM passed widely used quality controls, indicating reasonably Markovian dynamics (Supplementary Figs. S10, S11). However, the MSM suggested a MFPT of only ~10 μs for the opening process, which is tenfold lower than the value of 100 μs obtained by the linear kinetic model. This bias in the MSM is likely explained by insufficient sampling of the rate-liming loop-to-helix transition of the sensor serine described below. Hence, whereas the MSM and the free energy landscape presented here should be interpreted with care, the linear kinetic model from successful AS simulations provided a simple and numerically robust approximation to the Prp43 kinetics.

## Molecular switches of the opening transition

*Transition of the RecA2 β-hairpin and RNA backbone rotation.* As the first critical transition after removal of the ATP, the β-hairpin of RecA2 carried out a rapid rearrangement within the first nanoseconds of the AS simulations (Fig. 3a–c). The rearrangement involved a cleavage of the K403–U4 H-bond and a sudden drop of the K403–U3 distance from 0.9 nm to 0.4 nm (Fig. 3c). This rapid transition is an indicator of a strong tension in the RNA–RecA2 interface of the closed state, which was likely stabilized by electrostatic interactions of the ATP with RecA1 and RecA2. Hence, the tension was released in an instant after ATP release.

Cleavage of the K403–U4 H-bond allowed the formation of the K403–U3 H-bond after 180 ns of the cumulative simulation time, as required for sliding of RecA2 along the RNA by one base pair (Fig. 3c, 180 ns). To enable the formation of the K403–U3 H-bond, a rotation of the RNA backbone around the P–P axis between U3 and U4 was strictly required, thereby pointing the H-bond acceptor $O^{1A}$ of U3 towards K403 (Fig. 3c, black arrow). This RNA backbone rotation was, in turn, only enabled by the major opening step during the first 100 ns described above, which rendered RecA2 more flexible and provided the RNA with increased conformational freedom. For a view on the overall dynamics of the RNA rotation, we refer to Supplementary Movie S1. The successful shift of K403 from U4 to U3 left the U4 phosphate vacant, which enabled the U4 phosphate to rotate and to form a critical connection of U4 with the hook-loop and with T381 described below (Fig. 3i).

*Arginine finger R435 of RecA2 as anchor for the S387–G392 sensor loop.* Arginine fingers are a reoccurring motif of NTP-binding sites, where they are crucial for stabilizing the NTP hydrolysis complex[54–56]. In the closed conformation of Prp43, the arginine finger R435 of RecA2 is part of the ATP binding pocket and interacts with the β- and γ-phosphate of ATP, thereby bridging the two RecA-like domains (Fig. 1). The nearby residue T389 interacts with R435 and with the adenosine moiety of ATP (Fig. 1, Fig. 3d). After release of the ATP, the arginine finger shifted away from the RecA1 domain and broke the contact with T389 (Fig. 3e). Cleavage of the R435–T389 contact allowed increased flexibility of the loop S387–G392, to which we refer as "sensor loop" since it contains the sensor serine S387[36]. This led to an unfavorable steric clash of the sensor loop with the R153–L167 helix of RecA1, possibly contributing to a repulsion between the two RecA-like domains. At this stage, any further transition of the sensor loop towards its conformation in the open state was obstructed by the current H-bond of U5 with the hook-loop and with T381 (Fig. 3h).

*Upstream motion of the hook-loop and T381 from U5 to U4.* The hook-loop is an important anchor between the RecA2 domain and the RNA. At the beginning of the cycle in the closed state, G349 of the hook-loop and the nearby T381 form H-bonds with the U5 phosphate of the RNA (Fig. 3h). During the opening transition, these H-bonds shift by one nucleotide upstream from U5 to U4 (Fig. 3i). As illustrated in the G349–U4 and T381–U4 distances in Fig. 3j, k, the upstream motion of G349 and T381 occurred in two steps: the first step started at 180 ns and correlated with the RNA backbone rotation between U3 and U4 and the formation of the K403–U3 H-bond described above (Fig. 3b, c). The second step at 400 ns, accompanied by a rotation of U4, finalized the upstream motion of G349 and T381 (Fig. 3i). This transition left U5 vacant, henceforth allowing the H-bond formation by the sensor serine with U5.

*Loop-to-helix transition of sensor loop and binding of sensor serine S387 to U5.* Previous crystallographic data suggested that transitions of the "sensor serine" S387 are critical for RNA translocation[36]. After the cleavage of the H-bond between the arginine finger with T389 described above, the sensor serine S387 (as part of the sensor loop S387–G392) carried out a loop-to-helix transition, decreasing the S387–U5 distance by ≈7 Å (Fig. 3n, 100 ns, black arrow). However, only after the upstream motion of the β-hairpin from U4 to U3 (Fig. 3a–c, 180 ns) and of the hook-loop from U5 to U4 (Fig. 3h–k, 400 ns), the U5 phosphate was vacant, enabling the sensor serine to form a new H-bond with the RNA (Fig. 3m, n). The formation of the S387–U5 H-bond finalized the successful opening transition.

Notably, simulating the loop-to-helix transition of S387 with AS was challenging. Among 100 simulation of 100 ns, we observed only a single successful loop-to-helix transition, likely because such changes of secondary structure are slow in sterically tight environments (Supplementary Fig. S4). Hence, the loop-to-helix transition was a rate-limiting step of the opening transition after the release of ATP.

We found that the degree of opening, as given by the RecA1–RecA2 distance, is correlated with the loop-to-helix transition of sensor serine. Namely, a large RecA1–RecA2 distance of ~3.1 nm could be maintained only if the sensor loop was in the helical state (Fig. 3o). As an independent test for this correlation, we carried out two sets of non-equilibrium pulling simulations, in which we either increased the RecA1–RecA2 distance to enforce an Prp43 opening, or we decreased the S387–U5 distance to enforce a loop-to-helix transition of the sensor loop (Supplementary Fig. S5). After releasing the pull forces, Prp43 remained open only in the simulations with a successful loop-to-helix transition, whereas other simulations exhibited partial re-closure of the RecA1–RecA2 interface. These simulations confirm the findings from AS (Fig. 3o), namely that the loop-to-helix transition by the sensor loop is strictly required for a successful opening transition of Prp43.

*Hook-loop and serine-loop translocations are encoded in the RecA2 dynamics.* As described above, both the hook-loop and serine-loop transitions are strictly required for a successful Prp43 opening. Hence, we asked whether these dynamics are guided by the RecA2 interactions with the RNA, or whether they are intrinsically encoded in the dynamics of the RecA2 structure. To this end, we carried out an additional microsecond simulation of the isolated RecA2 domain and analyzed the intrinsic dynamics using principal component analysis (PCA; Fig. 4a). The PCA revealed that both the hook-loop and serine loop are highly dynamic in the isolated RecA2 domain, and the first PCA vector describes transitions as observed during the upstream sliding along the RNA. This finding suggests that the RecA2 structure

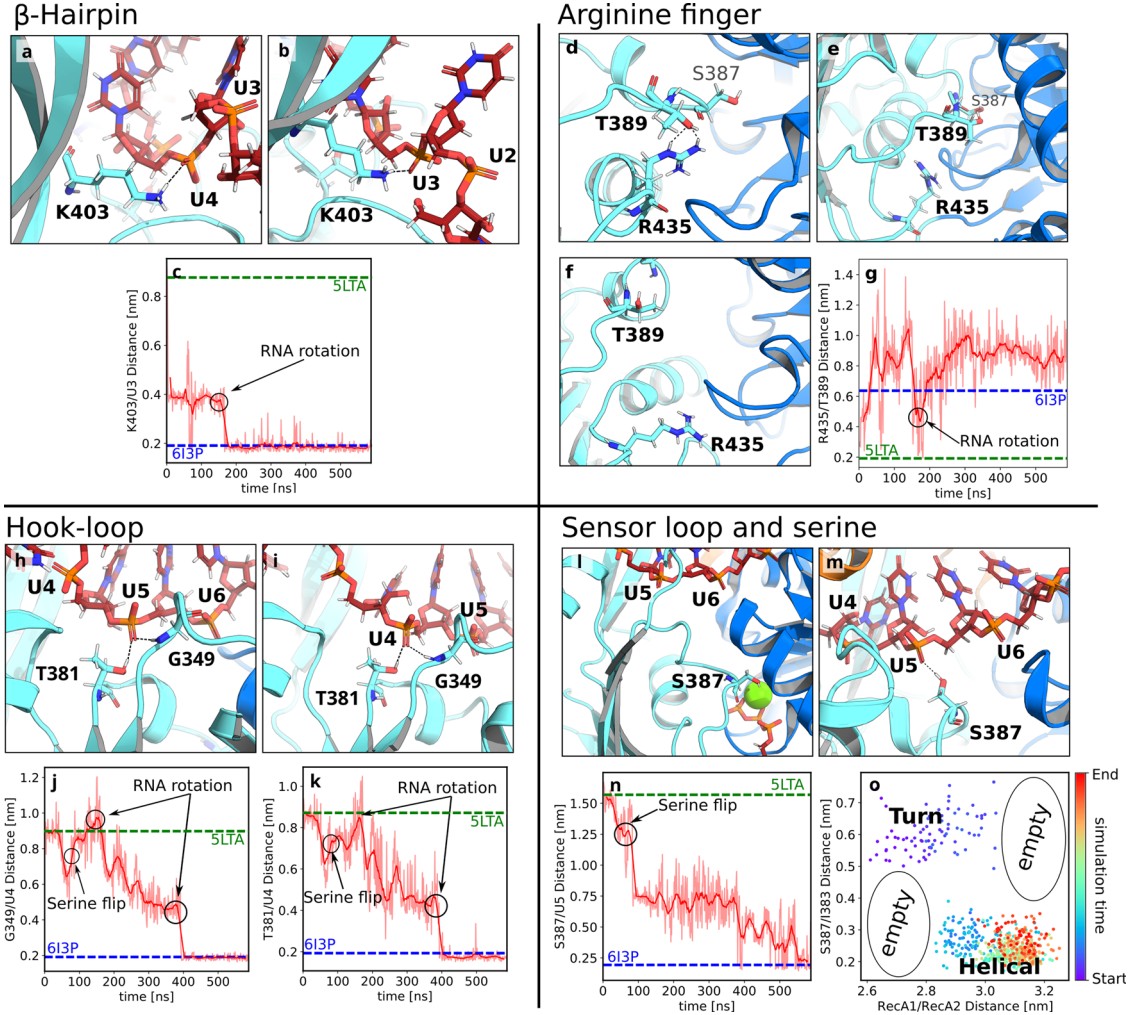

**Fig. 3 Molecular switches of the opening process.** Conformational transitions of (**a–c**) the β-hairpin, (**d–g**) the arginine finger, (**h–k**) the hook-loop, and (**l–o**) the sensor loop during Prp43 opening. (**a, d, h, l**) Molecular switches in the initial and (b/f/i/m) final frames of adaptive sampling of the open process. (**c, g, j, k, n**) In the distance plots, the time averages are drawn as dark red lines while the underlying data is drawn in pale red. Critical atomic distances that quantify the progression of the opening transition, where dashed lines indicate the distances in the 5LTA and 6I3P crystal structures. **a** View from the backside of Prp43: K403 of the β-hairpin hydrogen bound to U4 in closed conformation and (**b**) shifted from U4 to U3 in open conformation. **c** K403–U3 distance versus cumulative simulation time. **d** Closed RecA1–RecA2 interface with the Arginine finger R435 hydrogen bound to T389 and located on top of the P-loop. **e** Broken R435–T389 hydrogen bond after 5 ns. **f** Final conformation with distant R435 and T389 residues and with the arginine finger located underneath the P-loop. **g** R435–T389 distance versus cumulative simulation time. **h** Closed conformation, hook-loop bound to U5 via G349 and T381. **i** Open conformation, G349 and T381 shifted from U5 to U4. S387 formed an H-bond to U5 similar to the conformation in panel M. **j** G349–U4 distance and (**k**) T381–U4 versus cumulative simulation time. **l** Serine finger S387 bent towards the ATP in closed conformation and (**m**) pointing towards the RNA forming an H-bond with U5. **n** S387–U5 distance versus cumulative simulation time. **o** S387–I383 distance vs. RecA1–RecA2 distance, revealing a marked correlation between the sensor serine conformation (helical or turn) and Prp43 opening. Supplementary Fig. S12 shows the relative position of the zoom-in features to the overall structure.

and its intrinsic dynamics have been optimized for enabling the critical H-bonds shifts described above and, thereby, for RecA2 sliding along the RNA.

### Molecular switches of the closing transition

*Anchoring the arginine finger to the ATP.* We triggered the closing process by inserting ATP into the binding pocket of the RecA1 domain of the final frame of the opening simulations. The arginine finger R435 formed a stable contact with the β and γ phosphates of ATP within only 40 ns of a successful AS simulation (Fig. 5a–c). This rapid transition, driven by strong electrostatic R435–ATP interactions, anchored the RecA2 domain to the RecA1 domain via the ATP.

*Reverse transition of the sensor serine S387 is critical for closing the RecA1/RecA2 interface.* The sensor serine S387 played multiple critical roles during the closing process. S387 rapidly lost contact with U5 (Fig. 5d, f), thereby enabling U5 to form an H-bond with R153 of the RecA1 domain, as required for sliding the RNA along the RecA1 domain (Fig. 5h–j). In addition, the sensor loop carried out a helix-to-loop transition, thereby extending S387 underneath the R153–L167 helix of RecA1 where S387 interacts with a water molecule of the ATP–$Mg^{2+}$–water complex to stabilize the closed state. Notably, arrangement of water in the final closed state resembled the structure identified by crystallography[35,57] (Supplementary Fig. S14). The $Mg^{2+}$ ion was coordinated with three water molecules, thereby bridging interactions with nearby residues D218, E219, and S387. R432 and R435 interacted with water

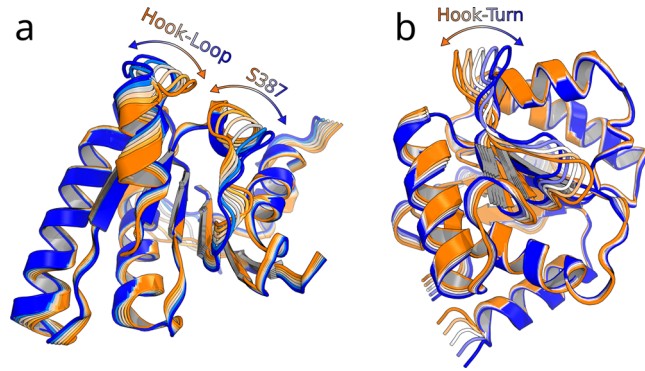

**Fig. 4 PCA of isolated RecA1 and RecA2. a** Motion along the first PCA vector of the isolated RecA2 domain visualized as 6 frames from blue to orange. The hook-loop and sensor serine exhibit the largest contributions to RecA2 fluctuations after excluding the highly mobile β-hairpin from PCA. **b** Motion along the first PCA vector of the isolated RecA1 domain. The hook-turn largely contributes to RecA1 fluctuations. Movies of these movements are shown in Supplementary Movies S3 and S4.

in the ATP pocked and directly with the phosphate moieties of the ATP, as observed by Tauchert et al.[35,57].

The helix-to-loop transition of the S387–G392 segment was strictly correlated with the closing of the RecA1–RecA2 interface, in line with the correlation between the loop-to-helix transition and the Prp43 opening described above (Figs. 5g, 3o). Namely, only after the transition to the loop state, as evident from an increased S387–I383 distance in Fig. 5g (red dots), a tight RecA1–RecA2 interface could form as indicated by a small RecA1–RecA2 distance. Additional support for this correlation was given by visual inspection of the simulations, confirming that the helix-to-loop transition followed by the extension of S387 beneath R152–L167 helix was required to enable tight packing of the RecA1/RecA2 interface without atomic clashes, as suggested previously from crystallographic data[35,36].

*RecA domain sliding along RNA.* Sliding of the RecA1 domain along the RNA involves the upstream motion of both the hook-turn and the R152–L167 helix, characterized by the transition of the backbone amine groups of R153 from U6 to U5, of T195 from U7 to U6, and of R180 from U7 to U6 (Fig. 5h–l). However, these transitions were initiated only after the sensor loop carried out the helix-to-loop transition, and they were completed only after the sensor serine bound to the ATP-Mg$^{2+}$-water complex at 400 ns (Fig. 5d–f). In addition to the backbone interactions of R153 and R180 with the RNA, the guanidinium moieties of these arginines frequently bound to the RNA backbone. The occasional release of these guanidinium–RNA H-bonds were required for a successful upstream motion. This may indicate that a fine balance in strength and number of RecA1–RNA versus RecA2–RNA H-bonds dictates the upstream motion of Prp43.

To test whether the hook-turn dynamics are intrinsically encoded into the RecA2 domain, we carried out a microsecond simulation of the isolated RecA1 domain, analogous to the simulations of the isolated RecA2 domain discussed above. PCA revealed that fluctuations of the RecA1 domain are dominated by fluctuations of the hook-turn, while the hook-turn transition during RecA1 sliding observed during AS occurred approximately along the first PCA vector (Fig. 4b). This analysis complements the PCA of the RecA2 domain described above, together suggesting the largest-scale fluctuation of both RecA-like domains are optimized for enabling RNA sliding.

Visual inspection of the closing trajectory revealed that RecA1–RecA2 interactions are not exclusively established via

the ATP. Instead, after sliding of RecA1 along the RNA, R152 of RecA1 frequently interacted with N382 of RecA2, while N382 occasionally interacted with the β-hairpin of RecA2 during the closing transition, which contributes to the tight packing of the RecA1–RecA2 interface and, thereby, to driving the upstream motion along the RNA (Fig. 5m–p). In summary, after the formation of the interface of RecA2 with the RecA1/ATP complex and the sliding of RecA1 along the RNA, the closing transition was completed.

## Discussion
All-atom simulations of complete conformational cycles of complex processes of enzymes, such as translocations, are still rare in the literature. Here we showed that AS simulations augmented with ST were capable of obtaining a complete RNA translocation cycle of the 80 kDa helicase Prp43. The simulations together with crystallographic studies suggest a translocation mechanism involving the domain rearrangements shown in Fig. 6a. Accordingly, ATP binding to apo Prp43 opens the RNA cleft, thereby allowing the binding of RNA (Fig. 6a(1.–3.))[35]. The hydrolysis of ATP enables the release of ADP and phosphate, while the phosphate ion is predominantly released via a tunnel on the backside of Prp43[58]. Removal of these negative charges leads to a spring-like conformational change in the ATP binding pocket, driving the movement of RecA2 by one RNA base upstream (Fig. 6a(4.–6.)). Binding of the next ATP triggers the closure of the RecA1–RecA2 interface by moving RecA1 along the RNA (Fig. 6a(7.–8.)). In our simulations, the CTD moved concertedly with the RecA2 domain (Fig. 6a(5.–8.)).

The simulations revealed that the key domain transitions are by no means characterized as diffusive center-of-mass movement of the entire domains. Instead, the large-scale domain dynamics are controlled by atomic-scale molecular switches that occur stochastically along orthogonal degrees of freedom of a highly rugged free energy landscape. Such dynamics have been studied extensively in the context of protein folding, yet much less for conformational cycles of motor enzymes such as Prp43[59,60]. Because the transitions of the molecular switches are interdependent, they occurred in a defined temporal order. Hence, our study highlights that large-scale domain motions of enzymes similar to Prp43 are controlled by atomic-scale molecular switches. This further implies that a mechanistic understanding of the enzyme kinetics, for instance involving the regulation by G-patches[61–64], requires identification of the molecular switches and understanding of their kinetics.

The overall domain displacement of Prp43 are compatible with an inchworm model[28]. However, an inchworm-like picture may imply that, during upstream motions, the RecA1 and RecA2 domains would fully detach from the RNA and re-bind one nucleotide upstream[36], which is not observed in our simulations (Fig. 6b). Instead, RecA1 and RecA2 remain bound to the RNA throughout the cycle and individually crawl along RNA by shifting protein–RNA H-bonds one-by-one, similar caterpillar walking. Hence, we suggest an inchworm/caterpillar model to describe both the relative domain displacements (inchworm) as well as the upstream movements of the individual RecA–RNA contacts (caterpillar; Fig. 6c).

In this study, we did not simulate ATP binding or hydrolysis or the release of ADP and phosphate, but instead focused on the domain motions initiated by the insertion and removal of ATP. Hence, the simulations demonstrate that the mere presence or absence of ATP is sufficient to trigger closing or opening of Prp43, respectively, thereby driving RNA translocation on the hundreds of microsecond timescale. This finding suggests that the energy from ATP hydrolysis is not required to generate a "kinetic

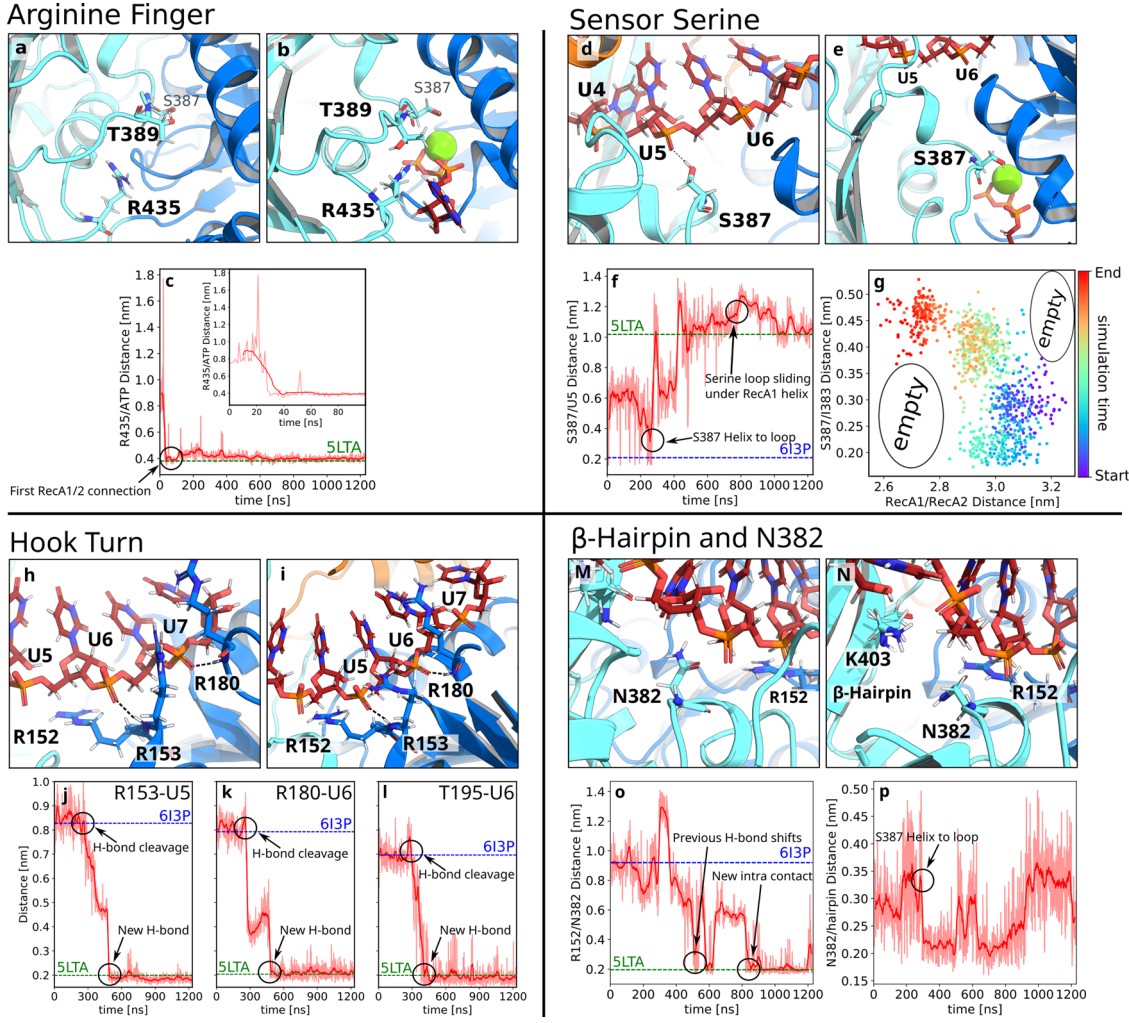

**Fig. 5 Molecular switches of the closing process.** Conformational transitions of (**a–c**) the arginine finger, (**d–g**) sensor serine, (**h–l**) RecA1–RNA interactions, and (**m–p**) β-hairpin during Prp43 closing. **a**, **d**, **h**, **m** Molecular switches in the initial and (**b**, **e**, **i**, **n**) final frames of adaptive sampling of the closing process. **c**, **f**, **j–l**, **o**, **p** In the distance plots, the time averages are drawn as dark red lines while the underlying data of each frame is drawn in pale red. Critical atomic distances that quantify the progression of the closing transition. Dashed lines indicate the distances in the 5LTA and 6I3P crystal structures. **a** Open RecA1/RecA2 interface with distant R435 and T389 residues. **b** Closed conformation with ATP in the binding pocket (stick representation), thereby bridging the closed RecA1/RecA2 interface. R435 is tightly bound to the phosphate groups of ATP. **c** R435/$C_\zeta$–ATP/O3B distance, revealing an R435–ATP H-bond formation early during the closing process. **d** Open conformation with the sensor serine S387 in the helical state and forming an H-bond with U5 of the RNA. **e** Closed conformation with S387 in the loop conformation, bound to the ATP–$Mg^{2+}$ complex, and reaching underneath the RecA1 R152-L167 helix. **f** R435–U5 distance during the closing process. **g** S387–I383 distance versus RecA1–RecA2 distance. The color indicates the cumulative simulation time. **h** Open conformation with R153 and R180 of RecA2 hydrogen bound to U6 and U7, respectively. **i** Closed conformation with R153 and R180 hydrogen bound to U5 and U6, respectively, shifted one nucleotide upstream relative to the open conformation. **j–l** R153–U5, R180–U6, and T195–U6 distances during the closing process. **m** Open conformation with N382 interacting with the β-hairpin. **n** Closed conformation with N382 interacting with R152, thereby connecting RecA2 and RecA1 domains. **o**, **p** R152–N382 distance and N382–β-hairpin distance. Supplementary Fig. S13 shows the relative position of the zoom-in features to the overall structure.

push" towards Prp43 opening but merely to allow dissociation of the ADP/phosphate products. Together, binding and release of ATP modulates the minima of the rugged free energy landscape, on which the domains move in a stochastic fashion.

Since the overall kinetics of RNA translocation are limited by transitions of molecular switches and not by diffusion of the overall domains, simulating such enzymatic cycles is challenging. Specifically, enhanced sampling techniques that merely enhance the diffusion of the complete domains or steer center-of-mass distances between domains are barely useful for such a system, since all the rate-limiting transitions occur in conformational space orthogonally to the domain center-of-mass distances[65]. Instead, methods such as milestoning[66] or, as used here, AS are

suitable for sampling large-scale domain motions of enzymes in rugged energy landscapes as it does not require the definition of reaction coordinates[41]. As a disadvantage, AS may bias the ensemble owing to the selection of the successful simulations, which may lead to over-representation of states with higher free energy; in this study, the enhanced sampling by the use of ST likely reduces such bias as ST accelerates the re-equilibration of the simulations at the beginning of the next AS round.

In addition, AS is suitable for estimating the kinetics of the overall pathway by collecting the rates between neighboring metastable states. For the successful sequence of transitions, using a simple linear kinetic model, we estimated the MFPTs of the opening and closing transitions in the order of 1 ms or 0.5 ms,

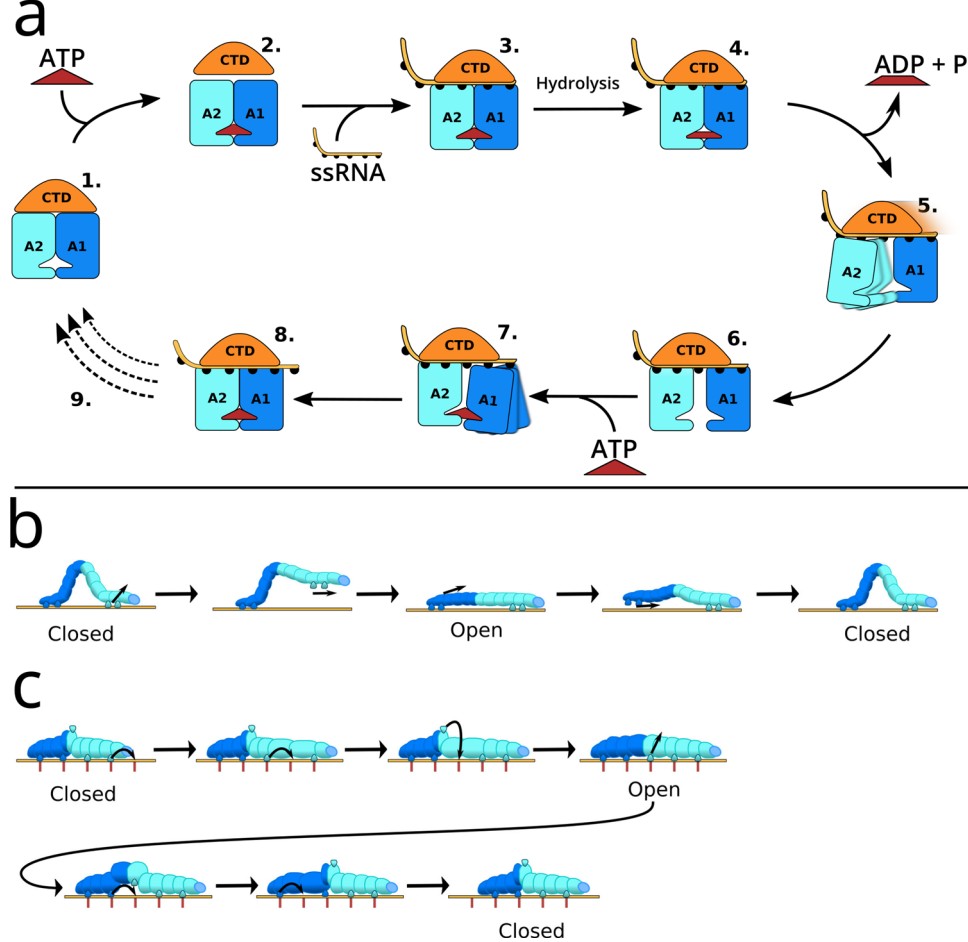

**Fig. 6 Mechanism and models of RNA translocation. a** Schematic hypothesis of the complete translocation cycle. 1. Apo structure. 2. ATP-binding triggers opening of the CTD/RecA interface, allowing 3. binding of ssRNA and formation of the Protein–ATP–RNA complex, represented by PDB ID 5LTA. 4. ADP bound state after ATP hydrolysis phosphate release. 5. RNA-bound state with open RecA interface and weak RecA2–RNA contacts. 6. Stable open structure, represented by PDB ID 6I3P. 7. ATP-binding triggers closure of the RecA interface and sliding of RecA1 along the RNA. 8. Protein–ATP–RNA complex translocated by one nucleotide relative to the 3. step. 9. Transition back to the apo-structure after finalizing multiple RNA translocations. **b** The classical inchworm model. **c** Proposed inchworm/caterpillar model to illustrate both the center-of-mass motion of the RecA-like domains (dark blue and cyan) and the crawling of RecA-like along the RNA. Hydrogen bond partners of protein and RNA are sketched as caterpillar legs and as red lines, respectively. Since the RecA-like domains bind the RNA with four or five H-bonds in the closed and open state, respectively, the caterpillar requires five legs. The central leg, modeling the sensor serine S387, carries out a rotation to bind the free nucleotide binding site before reaching the open state.

respectively (assuming tenfold accelerated rates owing to the use of ST). These values are in reasonable agreement with experimental data for other helicases[51–53]. Complementary, we constructed an MSM for the conformational cycle. While the MSM suggested a closing rate in reasonable agreement with the linear kinetic model, the MSM overestimated the opening rate, which we ascribe to insufficient sampling of the rate-limiting loop-to-helix transition of the sensor serine. Hence, for complex transitions as studied here, for which constructing a converged MSM remains challenging, the linear kinetic model provides a numerically robust and useful alternative.

Notably, by using AS, the computational cost for obtaining a conformational cycle was reduced only by a factor of approximately two compared to using few long simulations, primarily because the opening and closing processes occur down the gradient of the free energy landscape (Materials and Methods for details). Hence, a key advantage of AS for the present study was also the ability to trivially parallelize the simulations with commodity hardware, thereby drastically reducing the elapsed real time (or wall clock time) for completing the first conformational cycle. Another key for obtaining the Prp43 cycle with acceptable

computational costs was to augment AS simulations with ST. In line with the enhanced sampling of conformational transitions of small domains[39], we obtained significantly improved sampling of the enzyme dynamics (Supplementary Fig. S3), which rendered the simulations feasible. We expect that the combination of AS with ST will be useful for a wide range of future enzyme simulations.

In conclusion, we obtained a complete conformational cycle of the DEAH-box helicase Prp43 by combing adaptive sampling, simulated tempering, and crystallographic data. The cycle covered the opening and closure of the RecA1/RecA2 interface as well as upstream sliding of RecA1 and RecA2 along the RNA. By building kinetic models of Prp43 opening and closing, we estimated the mean first passage times being in the order of one millisecond, in line kinetic data found for other helicases. These findings suggest that simulations of millisecond conformational cycles of complex enzymes are feasible on commodity hardware using moderate computational effort.

The overall domain dynamics of Prp43 followed the well-established inchworm model of helicase function; however, because the RecA-like domains never detached from the RNA but

instead crawled along the RNA, by shifting protein–RNA hydrogen bonds one by one, we suggest a refined inchworm/caterpillar model to illustrate both the relative domain displacements and the upstream motions of RecA-like domains. Critically, the large-scale domain motions were controlled by a multi-step sequence of interdependent non-linear atomic-scale transitions, which were characterized in detail by the simulations. Such complex transitions are likely required to modulate the RecA1–RNA interaction relative to the RecA2–RNA interactions and, thereby, to guarantee the directionality of the process. We expect that enhanced sampling MD simulations, guided by structural data, will be a useful for revealing the molecular switches underlying energy-driven, directed domain motions in many other enzymes.

## Materials and methods

**Simulation setup of the opening process.** Molecular dynamics (MD) simulations of the opening process of the RNA translocation cycle were set up as follows with GROMACS 2019.5[67]. The initial structure of Prp43 from *C. thermophilum* was taken from the protein data bank (PDB ID 5LTA[35]), representing the complex of Prp43 with $U_7$-RNA and with the ATP analogue ADP-BeF$_3$. The ATP analogue and other inorganic molecules were removed from the structure, thereby modeling Prp43•$U_7$ after the dissociation of the hydrolyzed ATP. The structure was placed into a simulation box of a dodecahedron. The box was solvated with 35350 water molecules and neutralized with 9 potassium ions. Interactions of protein and RNA were described with the Amber14SB force field[68]. Water was modeled with the TIP3P model[69], and parameters for K$^+$ were taken from[70]. The energy of the system was minimized with the steepest descent algorithm. Then, the system was equilibrated for 100 ps with position restraints acting on the heavy atoms including RNA and Mg ($k = 1000$ kJ mol$^{-1}$ nm$^{-2}$).

The following parameters were used in all simulations. Electrostatic interactions were described with the particle-mesh Ewald method[71]. Dispersion interactions and short-range repulsion were described together with a Lennard-Jones potential with a cut-off at 1 nm. The temperature was controlled at 300 K using velocity-scaling[72], thereby coupling protein, RNA, Mg$^{2+}$, and ATP (if present) to one heat bath while coupling water and K$^+$ to a second heat bath ($\tau = 0.5$ ps). The pressure was controlled at 1 bar with the Parrinello-Rahman barostat ($\tau = 5$ ps)[73]. The md-vv integrator was used for simulated tempering (ST) simulations and the md integrator was used for all other simulations, both with an integration time step of 2 fs. The geometry of water molecules was constrained with SETTLE[74], while all other bonds were constrained with P-LINCS[75]. To accelerate the conformational sampling of the Prp43 cycle, we used adaptive sampling (AS) in combination with ST, as described in the following.

**Adaptive sampling protocol.** The Prp43 opening process required 9 rounds of AS, whereas the closing process required 11 rounds to complete the cycle of RNA translocation by one nucleotide. In each round, between 20 and 500 parallel simulations were carried out, which were started from the final conformation of the most successful simulation of the previous round. The parallel simulations were seeded with new random velocities taken from a Maxwell-Boltzmann distribution, thereby obtaining independent trajectories. The individual simulations were carried out between 10 and 100 ns (Supplementary Table S1). Thereby, the accumulated simulation time was 56.5 $\mu$s for the opening process and 40.0 $\mu$s for the closing process.

A simulation of an AS round was taken as most successful, if the final frame exhibited the highest similarity to a set of selected features of the features of the reference state (target values). The selected features are listed in Table 1. In this study, the analysis of each round was carried out by plotting the structural features in a heat table as shown in Supplementary Fig. S1. Here, the first and second line in the heat table list the feature values of the starting and the reference structure, respectively. Other lines show the feature values of the independent simulations. The colors indicate the similarity of a feature either to the starting structure (purple) or to the reference structure (yellow). Since it was difficult to weight different features in an automated manner, we selected the most successful simulation by human supervision. This way, from AS round to round, the simulations gradually approached the conformation of the reference state.

*Selected features for adaptive sampling.* The progression of the conformational transitions were monitored using the following structural features, where the atom names follow the PDB names: Distances between 1. the center of mass (COM) of RecA1 and RecA2, distances between the following pairs of atoms 2. G349-H and U5-O1P, 3. T381-OG1 and U5-O1P, 4. N382-OD1 and U4-2HO, 5. K403-NZ and U4-O1P, 6. E316-H and U4-O2P, 7. S387-HG and Mg, 8. S387-HG and U5-O1P, 9. G349-H and U4-O1P, 10. G349-H and U4-O2P, 11. T381-HG1 and U4-O1P, 12.

T381-HG1 and U4-O2P, 13. N382-OD1 and U3-2HO, 14. K403-NZ and U3-O1P, 15./16. the $\psi$ and $\phi$ angles of S387. More details are shown in Table 1.

The helicase Prp22 is homologous to Prp43 studied here, as demonstrated by sequence identity and similarity of 47% and 63%, respectively, according to a FASTA sequence alignment[76]. Furthermore, residues of the selected features are conserved among Prp22 and Prp43, except for G349 in Prp43 that is replaced by serine in Prp22. However, since G349 interacts with the RNA mostly via the protein backbone, this replacement has only a minor effect on the characterization of the opening transition. Hence, target values for the selected features were taken from the open Prp22 structure (PDB ID 6I3P[36]).

**Simulated tempering.** ST accelerates the conformational sampling of a system with a rough free energy landscape[38,39]. We used the ST implementation of GROMACS using a minimal temperature of 300 K and a maximum temperature of 348 K. The temperature difference between neighboring states was set to 4 K resulting in 23 states. Attempts for temperature transitions were carried out every 500 integration steps and accepted or rejected with the Metropolis algorithm. The initial weights were calculated using a preliminary simulated annealing simulations with the routine described by Park et al.[77]. The weights of the states were updated every 500 steps throughout the simulations using the Wang-Landau algorithm[78]. A representative example for the convergence of the weights and for the transitions among temperature states over time is shown in Supplementary Fig. S6.

To test whether ST enhances the conformational sampling relative to conventional simulations, we performed 10 conventional and 10 ST simulations of Prp43 with GROMACS 2020.2. The initial coordinates for these simulations were taken from PDB data bank (PDB ID 5D0U[35]), representing the Prp43/ADP complex. A non-RNA-loaded structure was chosen in order to ensure a higher flexibility of the RecA domains. ADP was removed from the system to trigger the opening process, and the system was solvated with 45800 water molecules and neutralized with 1 potassium ions. The energy of the system was minimized with the steepest descent algorithm. Then, the system was equilibrated for 100 ps with position restraints acting on the heavy atoms including RNA and Mg ($k = 1000$ kJ mol$^{-1}$ nm$^{-2}$).

Electrostatic interactions were described with the particle-mesh Ewald method[71]. Dispersion interactions and short-range repulsion were described together with a Lennard-Jones potential with a cut-off at 1 nm. The temperature was controlled at 300 K using velocity-scaling[72], thereby coupling protein, RNA, Mg$^{2+}$, and ATP (if present) to one heat bath while coupling water and K$^+$ to a second heat bath ($\tau = 0.5$ ps). The pressure was controlled at 1 bar with the Parrinello-Rahman barostat ($\tau = 5$ ps)[73]. The md-vv integrator was used for simulated tempering simulations with an integration time step of 2 fs. The geometry of water molecules was constrained with SETTLE[74], while all other bonds were constrained with P-LINCS[75].

Then, the progression of the RecA domain distances were analyzed after 300 ns. As shown in Supplementary Fig. S3, the partial opening of the RecA1/RecA2 interface is greatly accelerated in ST simulations as compared to conventional simulations.

**Simulation setup of the closing process.** MD simulations of the closing process of the RNA translocation cycle were set up as follow with GROMACS 2020.2. The initial coordinates of Prp43 were taken from the last successful AS simulation of the opening process, representing the protein–RNA complex in the open configuration (Fig. 2d, colored representation). The ATP–Mg$^{2+}$–water complex was inserted by first superimposing the RecA1 domain of the open complex onto the RecA1 domain of the crystal structure using a root mean-square deviation (RMSD) fit (PDB ID 5LTA[35]). Then, the ATP, Mg$^{2+}$ and water were positioned to match at the position of the ATP analogue and of the crystal water in the superimposed 5LTA structure. Parameters of the ATP were taken from Carlson et al.[79], translated into GROMACS format with the ACPYPE software[80].

The energy of the system was minimized with the steepest descent algorithm. Then, the system was equilibrated for 100 ps with position restraints acting on the heavy atoms including RNA and Mg ($k = 1000$ kJ mol$^{-1}$ nm$^{-2}$).

Electrostatic interactions were described with the particle-mesh Ewald method[71]. Dispersion interactions and short-range repulsion were described together with a Lennard–Jones potential with a cut-off at 1 nm. The temperature was controlled at 300 K using velocity-scaling[72], thereby coupling protein, RNA, Mg$^{2+}$, and ATP (if present) to one heat bath while coupling water and K$^+$ to a second heat bath ($\tau = 0.5$ ps). The pressure was controlled at 1 bar with the Parrinello–Rahman barostat ($\tau = 5$ ps)[73]. The md-vv integrator was used for simulated tempering simulations with an integration time step of 2 fs. The geometry of water molecules was constrained with SETTLE[74], while all other bonds were constrained with P-LINCS[75].

**Simulation setup of pulling simulations.** Non-equilibrium pulling simulations were set up as follows with GROMACS 2019.1. The initial coordinates were taken from the equilibrated crystal structure 5LTA with removed ATP. For the first four pulling simulations, the COM distance between RecA1 and RecA2 was chosen as reaction coordinate. A force constant of 10000 kJ/mol · nm and pull

rate of 0.005 nm/ns was applied for 100 ns to increase the RecA1–RecA2 distance. After 100 ns, each simulation was continued without restraints for an additional 100 ns. For the second four pulling simulations, the COM distance between S387 and U5 O1P atoms was chosen as reaction coordinate. A force constant of 5000 kJ/mol · nm and a pull rate of −0.012 nm/ns was applied for 100 ns to decrease the distance between S387 and RNA-U5 and, thereby, drive the loop-to-helix transition of the sensor loop. In two of these four pulling simulations, the loop-to-helix transition of the sensor loop triggered the opening of the RecA1–RecA2 interface, despite the fact that the RecA COM distance was not steered. After these two 100 ns simulations, each was continued by four independent 400 ns without restraints.

The energy of the system was minimized with the steepest descent algorithm. Then, the system was equilibrated for 100 ps with position restraints acting on the heavy atoms including RNA and Mg ($k = 1000$ kJ mol$^{-1}$nm$^{-2}$).

Electrostatic interactions were described with the particle-mesh Ewald method[71]. Dispersion interactions and short-range repulsion were described together with a Lennard–Jones potential with a cut-off at 1 nm. The temperature was controlled at 300 K using velocity-scaling[72], thereby coupling protein, RNA, Mg$^{2+}$, and ATP (if present) to one heat bath while coupling water and K$^+$ to a second heat bath ($\tau = 0.5$ ps). The pressure was controlled at 1 bar with the Parrinello–Rahman barostat ($\tau = 5$ ps)[73]. The md integrator was used with an integration time step of 2 fs. The geometry of water molecules was constrained with SETTLE[74], while all other bonds were constrained with P-LINCS[75].

**Simulations and PCA of isolated RecA1 and RecA2 Domains.** MD simulations of 1 $\mu$s for the two isolated RecA-like domains were set up as follows with GROMACS 2019.5. The initial coordinates of RecA1 (residues 97–273) and RecA2 (residues 274–458) were taken from the 5LTA structure[35]. Each domain was placed into a dodecahedral simulation box. The RecA1 box and the RecA2 box were filled with 6308 or 16592 waters molecules and neutralized with one chloride or three sodium ions, respectively. The energy of the system was minimized with the steepest descent algorithm. Then, the system was equilibrated for 100 ps with position restraints acting on the heavy atoms including RNA and Mg ($k = 1000$ kJ mol$^{-1}$ nm$^{-2}$).

Electrostatic interactions were described with the particle-mesh Ewald method[71]. Dispersion interactions and short-range repulsion were described together with a Lennard-Jones potential with a cut-off at 1 nm. The temperature was controlled at 300 K using velocity-scaling[72], thereby coupling protein, RNA, Mg$^{2+}$, and ATP (if present) to one heat bath while coupling water and K$^+$ to a second heat bath ($\tau = 0.5$ ps). The pressure was controlled at 1 bar with the Parrinello-Rahman barostat ($\tau = 5$ ps)[73]. The md-vv integrator was used for simulated tempering simulations with an integration time step of 2 fs. The geometry of water molecules was constrained with SETTLE[74], while all other bonds were constrained with P-LINCS[75].

PCA was performed to reveal large-scale motions in the isolated RecA-like domains. The GROMACS module *gmx covar* was used to calculate and to diagonalize to the covariance matrix. Here, the PCA was applied to the backbone atoms excluding the heavy fluctuating residues T252-N264 of the RecA1 domain and the residues (including the $\beta$-hairpin) T401-I421 of the RecA2 domain. The interpolation between the extreme projections of the free simulations onto the first PCA vector of RecA1 and RecA2 are shown in Fig. 4.

**Estimation of mean first passage times (MFPTs).** In this study, we focused on achieving successful opening and closing transitions of Prp43, rather than exhaustively sampling transitions between all long-living intermediate states. Consequently, we estimated only the order of magnitude of the MFPTs but do not aim towards a comprehensive kinetic network of Prp43 dynamics. Since we applied ST during AS simulation runs, we assume that the transition rates have been accelerated by one order of magnitude[39].

The MFPTs were estimated based on the formalism by Pande and Singhal[44], which involves the calculation of the transition rate matrix between long-living intermediate states. We constructed the transition matrix using the following assumptions: (i) Based on the AS simulations, we modeled both the opening and the closing process as seven transitions between $N = 7$ states. The forward rates $k_{n,n+1}$ were taken from the number of successful forward transitions per total simulation time of the AS round (Supplementary Table S1). These forward transitions are based on the features shown in Supplementary Fig. S7. (ii) Since the transitions of molecular switches were found to be highly interdependent, we assumed that the opening and closing transitions occurs predominantly via the linear sequence of transitions described in the Results section, suggesting that the rate matrix is tridiagonal ($k_{n,m} = 0$ if $|n - m| > 1$). (iii) Because both the opening transition (in the absence of ATP) and the closing transition (in presence of ATP) occur down the free energy landscape ($\Delta G < 0$), we assume that the backward rates $k_{n+1,n}$ are smaller than the respective forward rates $k_{n,n+1}$. To reveal the range of possible MFPTs, we considered the limiting cases where (a) the backward rates equal the forward rates ($k_{n,n+1} = k_{n+1,n}$), corresponding to $\Delta G = 0$, or (b) the backward rates equal 0.01 times the forward rates ($k_{n,n+1} = 100k_{n+1,n}$), corresponding to a marked downhill process with $\Delta G \ll 0$. These assumptions lead

to the following matrix equation:

$$\begin{bmatrix} p_{11} - 1 & p_{12} & 0 & \ldots & \ldots & 0 \\ p_{21} & p_{22} - 1 & p_{23} & 0 & \ldots & 0 \\ 0 & \ddots & \ddots & \ddots & \ldots & 0 \\ \vdots & & & & & \vdots \\ 0 & \ldots & 0 & p_{65} & p_{66} - 1 & p_{67} \\ 0 & \ldots & \ldots & 0 & 0 & 1 \end{bmatrix} \begin{bmatrix} x_1 \\ x_2 \\ \vdots \\ \vdots \\ \vdots \\ x_7 \end{bmatrix} = \begin{bmatrix} -\Delta t \\ -\Delta t \\ \vdots \\ \vdots \\ -\Delta t \\ 0 \end{bmatrix} \quad (1)$$

Here, $p_{n,m}$ denotes the probability of transitioning from state $n$ to state $m$ ($n, m = 1, \ldots, 7$) within the time delay $\Delta t = 1$ ns. Using other time delays between 0.1 ns and 20 ns led to nearly identical estimates for the MFTPs. The symbols $x_i$ denote the MFPTs from state $i$ to the final state $n = 7$. The probabilities obey $p_{n,n} + p_{n,n-1} + p_{n,n+1} = 1$. For the two limiting cases described above, we have $p_{n,n-1} = c \cdot p_{n,n+1}$ with $c = 1$ or $c = 0.01$, respectively.

The simulation data used to compute the transition rates are summarized in Supplementary Table S1 for the opening and the closing process. The table lists, for each round of AS, the overall simulation time of the round, the number of parallel simulations of the round, the number of successful forward transitions taken from all parallel simulations, and the computed forward rate $k_{n,n+1}$. According to Table S1 the forward rates for the opening are $k_{n,n+1} = (0.10, 0.10, 0.90, 0.13, 2.41, 0.45)$ $\mu$s$^{-1}$ and the forward rates for the closing are $k_{n,n+1} = (3.69, 0.29, 0.40, 1.00, 0.56, 0.06)$ $\mu$s$^{-1}$. The transitions were identified via the progression of structural features (Supplementary Fig. S7) and validated by extensive visual inspection of the simulations. In case that no successful forward transition occurred within an initial set of parallel simulations, a new set was simulated, thereby adding to the total simulation time of the AS round. The probabilities in Eq. (1) can be estimated for small rates via

$$p_{n,n+1} = \frac{N_{n,n+1}^{\mathrm{trans}}}{T_n} \Delta t = k_{n,n+1} \Delta t, \quad (2)$$

where $N_{n,n+1}^{\mathrm{trans}}$ is the number of forward transitions in AS round $n$, and $T_n$ is the overall simulation time of AS round $n$.

*Upper/lower bounds and order of magnitude estimates of MFPTs.* For the two limiting cases of (i) a nearly flat free energy landscape ($k_{n,n+1} = k_{n+1,n}$) or (ii) for a marked downhill process with $\Delta G \ll 0$ we obtained the following MFPTs. For case (i), we obtained for the opening $k_{\mathrm{opening}} = 141$ $\mu$s and closing process $k_{\mathrm{closing}} = 52$ $\mu$s, representing upper bounds of the opening and closing rates. For the marked downhill process, we obtain $k_{\mathrm{opening}} = 32$ $\mu$s and $k_{\mathrm{closing}} = 28$ $\mu$s. As a more realistic intermediate case of $k_{n,n+1} = 2.5k_{n+1,n}$, which translate into $\Delta G \approx -18$ kJ/mol, we obtain $k_{\mathrm{opening}} = 50$ $\mu$s and $k_{\mathrm{closing}} = 31$ $\mu$s. These values imply orders of magnitudes for the opening and closing rates of 100 $\mu$s and 50 $\mu$s (see Results).

*Increased computational efficiency due to adaptive sampling.* The cumulative invested simulation time from all AS runs for the opening and closing simulations were 56 $\mu$s and 40 $\mu$s, which are both in the order of magnitude of the MFPTs. Hence, main benefit of AS was not to largely reduce the cumulative computational cost for obtaining the cycle. Instead, these values imply that the benefit of AS was the ability to trivially parallelize and to monitor the progression of the simulations towards the target state. As a numerical example, for the Prp43 system, we obtained a simulation performance of 70 ns/day on a compute node with a six-core CPU and a Nvidia RTX 2080Ti GPU. Hence, simulating 150 $\mu$s for the conformational cycle (sum of MFPTs) on a single node would have required ~6 years. Without ST, this value would increase by another order of magnitude. By combing AS with ST, these unacceptable wall clock times were dramatically reduced using only commodity hardware, enabling the study of complex enzymatic cycles as described here.

**Construction of the MSM.** The construction of the MSM and its analysis was performed with PyEMMA 2.5.7[81] on all trajectories of the open and closing simulations obtained by the AS procedure. A sub-group of the features from the AS was chosen for the first step of the MSM construction, called the "featurization". In particular, we have chosen the following features: RecA1–RecA2 distance, T381–U5 distance, T381–U4 distance, G349–U5 distance, G349–U4 distance, K403–U4 distance, K403–U3 distance, E316–U4 distance, S387–U5 distance, S387–I383 distance, G349–U4 distance, and RMSD towards starting structure. The featurization coarse-grains the feature space of the protein dynamics and, thereby, simplifies the further processing by using only a reduced dimensionality, while keeping key dynamics of the feature space. Next, the time-lagged independent component analysis was performed to further reduce the dimensionality of the system[48,82,83], thus breaking the system down to the eight slowest collective motions as a linear combination of all other features. The lag-time used for our tICA decomposition was 5 ns. The resulting first two independent components were chosen for the projection of the kinetically-based model obtained by the MSM, because those two components describe the systems dynamics in a simplified fashion (Supplementary Fig. S10a).

The conformational microstates were generated from the tiCA components by the $K$-means clustering algorithm[84,85]. We have chosen the value of 100 for the number of cluster centers $K$. For an optimal number of cluster centers, we calculated the VAMP-2 score as a function of $K$ and checked for convergence as shown in Supplementary Fig. S10b. Another important parameter for an MSM is the lag-time $\tau$ of the Markov Model. Here, the resulting implied timescales (ITS) should converge as a function of lag-time to ensure Markovianity, as found for lag times larger than 20 ns (Supplementary Fig. S10c). Once the appropriate parameters were chosen, we computed the free energy landscape by re-weighting the trajectory frames with stationary probabilities from the MSM and projected the resulting free energies (Supplementary Fig. S9) on the first two tiCA components as suggested by the PyEMMA workflow. The PCCA+ algorithm was used to assign each microstate to a corresponding macrostate. In this study, we have chosen to describe the system with five macrostates. The corresponding Chapman–Kolmogorov test for validation is shown in Supplementary Fig. S11. The MFPTs between the macrostates were obtained using the PyEMMA functionality.

**Reporting summary**. Further information on research design is available in the Nature Portfolio Reporting Summary linked to this article.

## Data availability
Source data, concatenated trajectories of opening and closing from AS and the code for the MSM construction are available as repository at https://doi.org/10.5281/zenodo.7600325.

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

## Acknowledgements

The authors thank Florian Hamann and Ralf Ficner for insightful discussions and, in addition, Ralf Ficner for critically reading the manuscript. Financial support by the Deutsche Forschungsgemeinschaft is acknowledged (SFB 860/A16 and INST 256/539-1).

## Author contributions

R.A.H. performed simulations, analysis of data and figure creation. R.A.H. and J.S.H. interpreted data and wrote the manuscript. J.S.H. reviewed the manuscript and supervised the study.

## Funding

## Competing interests

The authors declare no competing interests.
