## [Peer Review File · Communications Biology]

Reviewers' comments:

Reviewer #1 (Remarks to the Author):

MANUSCRIPT: Continuous all-atom conformational cycle of a motor enzyme

AUTHORS: Robert A. Becker and Jochen S. Hub

DECISION: Accepted after major revisions

COMMENTARY:

In their submission to *Communications Biology*, Becker and Hub uncover the mechanism of the DEAH-box helicase enzyme opening and closing transitions. The authors used enhanced sampling techniques to record these processes. They provide a creative use of a flavor of adaptive sampling, where their adaptive sampling regime opts for simulated tempering simulations rather than classical molecular dynamics simulations. From their adaptive sampling-like regime, they stitch together, or concatenate, the trajectories which they categorized as crossing the open-to-close transition for their modeled helicase protein. After concatenating frames from "successful" trajectories, they present their data as time-series and mainly focus on structural events that have occurred. From these analyses, they describe a refinement to the traditional inchworm model for describing RNA translocation by their helicase enzyme.

This type of manuscript would be nice to see published and deserves merit. The use of an enhanced sampling technique within an adaptive sampling-like regime would be a nice addition to the literature; most people stick to classical simulations when performing adaptive sampling. This is especially welcoming for a manuscript such as this, where the primary results focus on the identification of a new mechanism. The authors dedicate a good amount of time being transparent about how they approached their adaptive sampling regime and study of this helicase enzyme, which sets a good standard for the enhanced sampling community. Overall, this is a manuscript that, upon publishing, would appeal to structural biologist and molecular dynamics practitioner audiences. However, there are some issues that need to be addressed.

MAJOR SCIENTIFIC ISSUES:

To be clear, I do not suspect any of these revisions to the manuscript to change the overall findings, or even the time-series figures shown in the Main Text. However, these suggested revisions are intended to uphold customary standards when it comes to adaptive sampling studies.

1. Per their MD Checklist, the authors stated that properties show equilibration or convergence based on their time-course analyses. The authors said: "Convergence of H-bonds shown in the figures of the main text [indicates convergence]." While this may be the case for traditional MD analyses, and certainly shows that the concatenated trajectory captures the process of open-to-close transitions, these plots do not reflect the convergence of sampling achieved by the study. While the authors did perform adaptive sampling, the method used herein is a very aggressive form of adaptive sampling known as supervised MD (SuMD). I direct the authors to these sources about SuMD:

<https://pubs.acs.org/doi/full/10.1021/acs.jcim.6b00499>

https://link.springer.com/protocol/10.1007/978-1-4939-8630-9_17

<https://pubs.acs.org/doi/full/10.1021/acs.jcim.9b01094>

While SuMD is a great approach for exploring a conformational process, the selection of starting states is very biased. As such, the sampling of rare states in high energy transitionary regions may outweigh that of otherwise metastable low energy states occupying the "true" phase space of their process of interest. Because of this limitation, is uncommon for researchers to publish work solely derived from SuMD results. However, Becker and Hub provide an interesting case where they have used simulated tempering to achieve a far greater exploration of the phase space.

To provide evidence that the sampling performed in this study is sufficient/converged, the authors should do the following:

A. Project their simulation data as a free energy landscape. The best simple way to do this would

be to calculate all the features presented in Figure 1 and store as a time-series vector per frame for each adaptive sampling trajectory. Then, using a dimensionality reduction method, like time-lagged independent component analysis (tICA), project the dimensionally-reduced data onto time-lagged independent components one and two (tIC1 versus tIC2). Several papers by the Pande and Noe groups have implemented tICA in their dimensionality reduction approaches; here is an example (<https://pubs.acs.org/doi/10.1021/ct300878a>). Other dimensionality reduction approaches have been used as well (<https://pubs.rsc.org/en/content/articlehtml/2022/ra/d2ra03660f>).

B. A tICA/reduced landscape can be generated using the pyEMMA software (<http://www.emma-project.org/latest/>) or the current version of MSMBuilder (<https://github.com/msmbuilder/msmbuilder2022>). Most manuscripts implementing adaptive sampling tend to show a free energy landscape. Becker and Hub can include theirs in the SI of this manuscript.

C. If the sampling is converged, then the metastable states observed in the tICA landscape would be fully connected. If not, then additional sampling would be required by the authors.

D. The authors should identify what the metastable states represent on these plots. Where are the open- and closed-like states? How strictly do they follow the values presented in Table 1?

At face value, generation of a raw tICA landscape would offer a glimpse as to whether the sampling the authors have performed is sufficient/converged. If the authors find that their data is not converged, they can select new seed simulations. See also Point 3 under OTHER COMMENTS about measuring extent of sampling bias.

2. In this manuscript, Becker and Hub estimate the mean first passage time (MFPT) for the successful opening and closing transitions of their helicase. While the exact approach used is a little outdated compared to making a Markov state model, the authors use the formalism provided by Vijay Pande's 2005 work to essentially construct their own transition probability matrix (<https://aip.scitation.org/doi/pdf/10.1063/1.2116947>). Becker and Hub state that "the forward rates were taken from the number of successful forward transitions per total simulation time of the AS round", as they show in Supplemental Table 1. Additionally, the authors stated that they selected a lag time of $dt = 1$ ns. Overall, I have a few concerns with this approach:

A. The code used to perform this calculation should be included in the Zenodo repository.

B. It is difficult to follow exactly how these forward/reverse transitions were calculated when the data in Supplemental Table 1 only includes observables and whether they are open-like or closed-like. Supplemental Table 1 does not indicate the number of transitions achieved per trajectory besides from qualitative observations demonstrated by a colorbar with an unknown scaling.

C. The authors have only inspected their data through visual inspection rather than as a free energy landscape, so it is unclear whether their sampling is converged. If the sampling is not converged, then it is difficult to say how accurate are these estimated MFPT calculations. Without convergence and incorporating the aggregate data into a master kinetic framework, the underlying ergodic assumptions of data obtained from adaptive sampling approximating Markovianity do not necessarily hold.

D. The authors did not justify their selection of the lag time. Ideally, the selected lag time should be suggestive of a converged implied time scale. A converged, or constant, implied timescale would suggest that the probability distribution used does indeed accurately approximate the underlying dynamics for the modeled system (see this webpage).

E. Because the selection of the 1 ns lag time seems arbitrary, I am not sure how trustworthy are these kinetics results.

F. The authors must demonstrate a rationale as to why they chose a 1 ns lag time. Ideally, the authors should optimize and validate a Markov state model using pyEMMA or MSMBuilder and show an implied timescale plot where they would select an optimal lag time for MFPT calculations. If the lag time differs, the authors should redo their MFPT calculations with the updated lag time.

The authors have a great opportunity here for generating a Markov state model (MSM) from adaptively sampled data obtained using an enhanced sampling technique. This could be a great example case study for MD practitioners, regardless of their interest in helicase.

OTHER COMMENTS

3. The authors should explain the aggressiveness of their adaptive sampling regime, and how their selection strategy does introduce increased bias into their simulations. Specifically, they should point out that their adaptive sampling strategy employs Supervised MD (SuMD). However, they should show that the extent of their sampling and its incorporation into a master kinetic framework (like a Markov state model) could remediate the bias introduced by seed state selection.

a. Extent of reweighting achieved by the Markov state model can be determined by showing that there is a linear relationship between the raw and MSM counts within the data. Basically, the log raw probability that a frame belongs to a cluster number should be roughly the same as the same frame's MSM-weighted probability of belonging to that same cluster. This type of analysis, as well as general MSM construction, is very reliant on good state discretization (feature-based clustering). This type of analysis has been observed in the following studies:

- i. <https://www.nature.com/articles/s41557-018-0077-9#Sec12>, Supplemental Figure 2b
- ii. [https://www.jbc.org/article/S0021-9258\(22\)00204-6/fulltext](https://www.jbc.org/article/S0021-9258(22)00204-6/fulltext), Supplemental Figure 21B
- iii. <https://www.biorxiv.org/content/10.1101/2022.10.12.511964v1>, Supplemental Figure 11

4. Concatenating the "successful" trajectories from the adaptive sampling regime is a really creative way to capture a single trajectory from the aggregate dataset, as well as make the results more appealing to an experimental audience. However, Figures 3 and 5 are extremely busy and have layouts which do not feel very intuitive at first glance. Is there a way these panels can be reformatted to make them easier to read? Aside from this, the time series from Figure 3 and Figure 5 could benefit from additional labels at critical events. For example, there is discussion about hydrogen bond rearrangements for some of these panels but it is hard to compare the manuscript main text to these figures; it is easy to get lost when looking at all the panels.

5. The snapshots and protein figures in this manuscript have been rendered very nicely. If needed for Figure 3 and 5 improvement, it could be a nice addition to provide expanded snapshots of these individual events with reference to the entire protein structure. These snapshots could be added to the SI so that these events can be structurally contextualized with the rest of the protein.

6. The SI for this manuscript submission was botched somehow. All the SI figures are embedded into the pdf, but none of the figures were labeled with titles or captions. Instead, I had to open peripheral files in a text editor and compare it to the order of the figures. Additionally, this made it very difficult to properly interpret what was going on during the MD movie files. Hopefully this can be fixed for the resubmission.

7. Would it be worthwhile to make mp4 movie files capturing the entire concatenated trajectories that were made from the adaptive sampling? This could add to the structural emphasis of this study.

8. The authors used a very strong restraint force during their very short equilibration runs (1000 kJ/mol*nm² for a 100 ps run). Are there other studies which use such a short equilibration simulation length? Would it not have been worthwhile to perform an equilibration without positional restraints?

9. There are a few typos in the manuscript, as well as inconsistent references to figures. For instance, sometimes figures are referenced as "Figure X", whereas other times the same figure is referenced as "figure X". All figures and tables should be referenced with capital (uppercase) letters, including Supplemental Figures (or at least some consistency should be maintained).

Reviewer #2 (Remarks to the Author):

This manuscript reports an interesting study on the molecular dynamics simulations of a DEAH-box helicase, which is an exemplary molecular motor protein. By combining simulated tempering (ST) and adaptive sampling (AS), the authors characterized the millisecond-scale conformational transitions between the open and closed states of the motor domains (RecA1 & RecA2) of Prp43. Interestingly, they found that loop-to-helix (helix-to-loop) transition of the sensor loop is a rate-limiting process for the opening (closing) of the RecA1-RecA2 domain interface. The all-atom simulations reveal the motions of key motifs that fit well with established structural models. In general, the manuscript is well-written and interesting to the helicase research community. However, the following issues should be addressed before the manuscript can be accepted for publication in *Commun. Biol.*

1. The author should calculate the free energy profiles projected along the structural features, such as a 2D free energy profile obtained using RecA1-RecA2 and S387-I383 distances shown in Fig. 3O and 5G. As each round of adaptive sampling has many independent simulated tempering simulations, the weighted histogram analysis method (JCTC, 2007, 3, 26-41) can be applied to obtain the free energy profiles. This analysis is important to locate intermediate states and check the convergence of the AS protocol.
2. The convergence of the AS protocol should be demonstrated. For example, Ref. 40 checked the AS convergence by calculating a key observable (binding free energy) over multiple iterations.
3. In the method section of mean first passage time (MFPT) estimation, it is unclear how the N=8 states were obtained (such as coordinates used for clustering). What are the features of each state (e.g., RecA1-RecA2 distance, sensor loop, hook-loop, arginine finger)? The MFPT calculations assumed the non-neighboring intermediate states have no transitions. This assumption can be verified after assigning each frame of a trajectory to a specific state.
4. For the opening simulation, the target values of the structural features were taken from the open structure of Prp22, a homolog to Prp43. How similar are Prp22 and Prp43 in terms of structure and sequence? Sequence alignment should be performed to check the similarity between Prp22 and Prp43. Are the residues used for the structural features conserved in the two helicases?
5. The polarity of Prp43 should be discussed in the manuscript. It is known that DEAH-box proteins translocate in the 3'→5' direction. Are the simulations able to illustrate the molecular basis for this polarity?

Minor issues:

1. The RMSD values from the target states should be measured to monitor how far the system is from the target at the end of each trajectory.
2. In the last paragraph of Discussion, a computational study (JACS, 2015, 137, 3031-3040) of an RNA helicase should be mentioned, as it employed pathway sampling methods (the string method and milestoning) to study the motor translocation. The MFPT calculation by milestoning, which is rather rigorous, revealed a 0.11-millisecond conformational transition triggered by ATP hydrolysis product release.
3. The authors might want to be more careful when comparing the calculated MFPTs to the experimentally measured translocation speed. It has been shown for other helicase systems that phosphate or ADP release is the rate limiting process for the functional cycle. The powerstroke (conformational change) can be orders of magnitude faster than the hydrolysis product release process.
4. The last paragraph of Methods talked about the increased computational efficiency. How many days did it take to complete the AS protocol for opening and closing transitions (with trivial parallelization and multiple GPUs)?

5. It would be great to make a movie to show the details of the helix-to-loop/loop-to-helix transitions of the sensor loop (zoom-in view).

6. - Table 2 is missing.

- In the first paragraph of Page 23, "56.500 ns" and "40.000 ns" should be "56.5 μ s" and "40.0 μ s", respectively.

- Need captions for the supplement figures.

Reviewer #3 (Remarks to the Author):

Summary: Prp43 is a DEAH box helicase involved in the translocation of prokaryotic single stranded RNA (ssRNA). Although the mechanism of translocation in Prp43 is somewhat understood through their X-ray crystal structures, the actual sequence of intermediate events is still unknown. Through the present work, the authors utilized the concerted power of Adaptive Sampling (AS) and Simulated Tempering (ST) molecular dynamics (MD) simulations to capture atomistic details of the conformational cycle of Prp43. While ST helps in lowering the energetic barriers using higher temperatures, AS help in conformational sampling of rate-limiting steps as well as parallelize such sampling, thereby gaining computational efficiency over conventional MD. The authors used crystallographic data as start and end points, as well as crystallographic features to guide the course of the simulations. The work demonstrates that combination of the two enhanced sampling methods could help interpret subtle but kinetically important events, saving folds of computational times over conventional MD. Additionally, the detailed atomistic study of Prp43 helicases suggested a modification to the conventional inchworm motion of the RecA domain over RNA. The authors suggest that while the center of mass of RecA follows an inchworm like motion over RNA, the domains rather follow a caterpillar like crawling movement. Thus, the authors demonstrated that large scale conformational changes are guided by small scale atomistic events, which could be captured using a combination of the AS and ST methods.

Significance:

To date, multiple authors have worked on using all-atom simulations to decipher the dynamics of motor proteins¹⁻³. Additionally, the crystal structures of the RNA bound DEAH box helicases such as Prp43 were only recently resolved, and the conformational cycle was outlined⁴. Despite this, the intermediate steps in the conformational cycle of such helicases is still unknown. Given the size of the helicase-RNA-ATP/ADP complexes, simulation of conformational changes at an atomic scale incurs unrealistic computational cost. To address this, the authors suggest utilizing a combination of two enhanced sampling methods such as AS and ST, to overcome the energy barriers in the conformational cycle. As communicated through the article, the significance of the work is threefold:

- a. Utilization of two enhanced sampling methods such as Adaptive Sampling (AS) and Simulated Tempering (ST) to enhance computational efficiency in sampling rare events in the complex conformational cycle of a large motor protein.
- b. Importance of all-atom simulations in deciphering the key steps in conformational cycle of large biomolecular complexes, as illustrated by the role of the key biomolecular residues.
- c. Essential residues labelled as "molecular switches" are identified through the present work.

Although the roles of such switches are well identified through X-ray crystal structures, the actual sequence of events is not understood in detail. Such details could be used in the further study of DEAH box helicases.

The authors have presented a clear and detailed description of the work, including the application of the two methods and explanation of the corresponding results. The justification of choosing the parametric features for the simulations has been discussed and compared with the experimental start and endpoints for validation. Overall, the work presented by the authors is entirely novel and could be applied to study conformational dynamics of large biomolecular systems, and also help understand molecular mechanisms of motor proteins.

Comments:

The reviewer has the following comments related to the article:

1.Modification of the title: The title "Continuous all-atom conformational cycle of a motor enzyme" suggests that the main purpose of the article is achieving an all-atom simulation of a motor enzyme. This is although somewhat true, does not fully bring out the potential of the work communicated through the article. The reviewer suggests that the title of the article be modified to capture the interest of readers who would want to apply such methods on their systems and to address those who would be interested in studying Prp43.

2.Labelling residues in videos for clarity to the discussions:

The reviewer thinks that major residues should be labelled in the videos for clarity to the readers. For example, the authors have clearly described the role of K403 H-bond interactions with U4 and U3 as the molecular switch, through discussions in text and images in Figure 3 (A-C). However, labels of such residues in the video are missing. Labelling these residues/side chains in the videos will allow the readers to coherently relate the text and figure 3 to video 1.

3.Role of water molecules:

a. The role of a water molecule in engaging the arginine finger (R435) has been mentioned by Tauchert et al. (2017), who resolved the crystal structure of Prp43-ADP-BeF3-U7-RNA complex 4. The reviewer is curious if the authors found any such water mediated mechanism related to the anchoring of the arginine finger.

b. Additionally, although the authors mentioned the role of a water molecule in the stabilization of the closed state, by engaging S387 of RecA1, the respective figure 5E does not depict the water molecule.

"where S387 interacts with a water molecule of the ATP-Mg²⁺-water complex to stabilize the closed state (Fig. 5E)."

4. Role of buffer solution:

The conformational dynamics of nucleic acids are well known to be influenced by the presence of buffer solutions such as KCl and/or MgCl₂. The importance of metal cations and buffer solutions have been found to be important in the activity of helicases via multiple studies 5-7.

Moreover, the experimental conditions included the addition of 150 mM KCl and 3 mM MgCl₂ at a pH of 7.4 4.

Despite this, the method section for the preparation of the system only includes addition of neutralizing ions, but not buffer:

a. The reviewer is curious if this is only a typo, and the authors did add a buffer during solvation. If otherwise, what was the rationale behind omitting the buffer solution(s)?

b. Is it possible that the observations with the electrostatic switches might have been different if buffer was present? Could the "crawling" instead of a typical "inchworm" motion be related to the absence of a buffer?

5. Flowcharts for the methods of AS and ST:

Although the authors have described the methods of AS and ST very clearly, a flowchart of the various steps of the two methods will allow the reader to comprehend better. These flowcharts could be added as supplementary information.

References mentioned in comments:

1. Hwang W, Lang MJ, Karplus M. Kinesin motility is driven by subdomain dynamics. *Elife*. 2017;6.
2. Shi, X.X., Wang, P.Y., Chen, H. and Xie, P., 2021. Studies of conformational changes of tubulin induced by interaction with kinesin using atomistic molecular dynamics simulations. *International journal of molecular sciences*, 22(13), p.6709.
3. Flechsig, H., Popp, D. and Mikhailov, A.S., 2011. In silico investigation of conformational motions in superfamily 2 helicase proteins. *PLoS One*, 6(7), p.e21809.
4. Tauchert, M.J., Fourmann, J.B., Lührmann, R. and Ficner, R., 2017. Structural insights into the mechanism of the DEAH-box RNA helicase Prp43. *Elife*, 6.
5. Brennan, C.A., Steinmetz, E.J., Spear, P. and Platt, T., 1990. Specificity and efficiency of rho-factor helicase activity depends on magnesium concentration and energy coupling to NTP hydrolysis. *Journal of Biological Chemistry*, 265(10), pp.5440-5447.
6. Cao, X., Li, Y., Jin, X., Li, Y., Guo, F. and Jin, T., 2016. Molecular mechanism of divalent-metal-induced activation of NS3 helicase and insights into Zika virus inhibitor design. *Nucleic acids research*, p.gkw941.
7. Frick, D.N., Banik, S. and Rypma, R.S., 2007. Role of divalent metal cations in ATP hydrolysis catalyzed by the hepatitis C virus NS3 helicase: magnesium provides a bridge for ATP to fuel unwinding. *Journal of molecular biology*, 365(4), pp.1017-1032.

Reviewer #1 (Remarks to the Author):

MANUSCRIPT: Continuous all-atom conformational cycle of a motor enzyme

AUTHORS: Robert A. Becker and Jochen S. Hub

DECISION: Accepted after major revisions

COMMENTARY:

In their submission to *Communications Biology*, Becker and Hub uncover the mechanism of the DEAH-box helicase enzyme opening and closing transitions. The authors used enhanced sampling techniques to record these processes. They provide a creative use of a flavor of adaptive sampling, where their adaptive sampling regime opts for simulated tempering simulations rather than classical molecular dynamics simulations. From their adaptive sampling-like regime, they stitch together, or concatenate, the trajectories which they categorized as crossing the open-to-close transition for their modeled helicase protein. After concatenating frames from “successful” trajectories, they present their data as time-series and mainly focus on structural events that have occurred. From these analyses, they describe a refinement to the traditional inchworm model for describing RNA translocation by their helicase enzyme.

This type of manuscript would be nice to see published and deserves merit. The use of an enhanced sampling technique within an adaptive sampling-like regime would be a nice addition to the literature; most people stick to classical simulations when performing adaptive sampling. This is especially welcoming for a manuscript such as this, where the primary results focus on the identification of a new mechanism. The authors dedicate a good amount of time being transparent about how they approached their adaptive sampling regime and study of this helicase enzyme, which sets a good standard for the enhanced sampling community. Overall, this is a manuscript that, upon publishing, would appeal to structural biologist and molecular dynamics practitioner audiences. However, there are some issues that need to be addressed.

REPLY: We thank the reviewer for these positive remarks.

MAJOR SCIENTIFIC ISSUES:

To be clear, I do not suspect any of these revisions to the manuscript to change the overall findings, or even the time-series figures shown in the Main Text. However, these suggested revisions are intended to uphold customary standards when it comes to adaptive sampling studies.

1. Per their MD Checklist, the authors stated that properties show equilibration or convergence based on their time-course analyses. The authors said: “Convergence of H-bonds shown in the figures of the main text [indicates convergence].” While this may be the case for traditional MD analyses, and certainly shows that the concatenated trajectory captures the process of open-to-close transitions, these plots do not reflect the convergence of sampling achieved by the study. While the authors did perform adaptive sampling, the method used herein is a very aggressive form of adaptive sampling known as supervised MD (SuMD). I direct the authors to these sources about SuMD:

<https://pubs.acs.org/doi/full/10.1021/acs.jcim.6b00499>

https://link.springer.com/protocol/10.1007/978-1-4939-8630-9_17

<https://pubs.acs.org/doi/full/10.1021/acs.jcim.9b01094>

While SuMD is a great approach for exploring a conformational process, the selection of starting states is very biased. As such, the sampling of rare states in high energy transitional regions may outweigh

that of otherwise metastable low energy states occupying the “true” phase space of their process of interest. Because of this limitation, is uncommon for researchers to publish work solely derived from SuMD results. However, Becker and Hub provide an interesting case where they have used simulated tempering to achieve a far greater exploration of the phase space.

REPLY: Thank for pointing out the relation to the SuMD method, which is indeed similar to the adaptive sampling protocol used here, and for pointing towards the relevance of ST for accelerated equilibration to avoid the bias. We now write:

"AS is similar to the supervised molecular dynamics (SuMD) approach, which has been used to enhance the sampling of ligand binding simulations. \cite{sabbadin2018,deganutti2020} In contrast to frequently used autonomous supervising algorithms, we have chosen the “successful” AS simulations by human supervision by carefully investigating the features after each round of the AS (Supplementary Fig. S1)."

In addition, we added to the Discussion:

"As a disadvantage, AS may bias the ensemble owing to the selection of the successful simulations, which may lead to over-representation of states with higher free energy; in this study, the enhanced sampling by the use of ST likely reduces such bias as ST accelerates the re-equilibration of the simulations at the beginning of the next AS round."

REVIEWER:

To provide evidence that the sampling performed in this study is sufficient/converged, the authors should do the following:

A. Project their simulation data as a free energy landscape. The best simple way to do this would be to calculate all the features presented in Figure 1 and store as a time-series vector per frame for each adaptive sampling trajectory. Then, using a dimensionality reduction method, like time-lagged independent component analysis (tICA), project the dimensionally-reduced data onto time-lagged independent components one and two (tIC1 versus tIC2). Several papers by the Pande and Noe groups have implemented tICA in their dimensionality reduction approaches; here is an example (<https://pubs.acs.org/doi/10.1021/ct300878a>). Other dimensionality reduction approaches have been used as well (<https://pubs.rsc.org/en/content/articlehtml/2022/ra/d2ra03660f>).

REPLY: This comment and the major comments below are related and address the kinetic model, the free energy landscape, and the convergence. Hence, we provide a joint reply to these major remarks presented below.

REVIEWER:

B. A tICA/reduced landscape can be generated using the PyEMMA software (<http://www.emma-project.org/latest/>) or the current version of MSMBuilder

(<https://github.com/msmbuilder/msmbuilder2022>). Most manuscripts implementing adaptive sampling tend to show a free energy landscape. Becker and Hub can include theirs in the SI of this manuscript.

C. If the sampling is converged, then the metastable states observed in the tICA landscape would be fully connected. If not, then additional sampling would be required by the authors.

D. The authors should identify what the metastable states represent on these plots. Where are the open- and closed-like states? How strictly do they follow the values presented in Table 1?

At face value, generation of a raw tICA landscape would offer a glimpse as to whether the sampling the authors have performed is sufficient/converged. If the authors find that their data is not converged, they can select new seed simulations. See also Point 3 under OTHER COMMENTS about measuring extent of sampling bias.

2. In this manuscript, Becker and Hub estimate the mean first passage time (MFPT) for the successful opening and closing transitions of their helicase. While the exact approach used is a little outdated compared to making a Markov state model, the authors use the formalism provided by Vijay Pande's 2005 work to essentially construct their own transition probability matrix (<https://aip.scitation.org/doi/pdf/10.1063/1.2116947>). Becker and Hub state that "the forward rates were taken from the number of successful forward transitions per total simulation time of the AS round", as they show in Supplemental Table 1. Additionally, the authors stated that they selected a lag time of $\Delta t = 1$ ns. Overall, I have a few concerns with this approach:

A. The code used to perform this calculation should be included in the Zenodo repository.

B. It is difficult to follow exactly how these forward/reverse transitions were calculated when the data in Supplemental Table 1 only includes observables and whether they are open-like or closed-like. Supplemental Table 1 does not indicate the number of transitions achieved per trajectory besides from qualitative observations demonstrated by a colorbar with an unknown scaling.

REPLY:

We fully agree. The transition rates are now included in the text of the Methods and in Supplementary Table 1. Regarding the "code", since obtaining the MFPTs x_i in Eq. 1 simply requires the solution of a set of linear equations, we did not use a specific code but merely used a web service solver for linear coupled equations.

REVIEWER:

C. The authors have only inspected their data through visual inspection rather than as a free energy landscape, so it is unclear whether their sampling is converged. If the sampling is not converged, then it is difficult to say how accurate are these estimated MFPT calculations. Without convergence and incorporating the aggregate data into a master kinetic framework, the underlying ergodic assumptions of data obtained from adaptive sampling approximating Markovianity do not necessarily hold.

REPLY:

Please see our reply below

D. The authors did not justify their selection of the lag time. Ideally, the selected lag time should be suggestive of a converged implied time scale. A converged, or constant, implied timescale would suggest that the probability distribution used does indeed accurately approximate the underlying dynamics for the modeled system (see this webpage).

E. Because the selection of the 1 ns lag time seems arbitrary, I am not sure how trustworthy are these kinetics results.

F. The authors must demonstrate a rationale as to why they chose a 1 ns lag time. Ideally, the authors should optimize and validate a Markov state model using pyEMMA or MSMBuilder and show an implied timescale plot where they would select an optimal lag time for MFPT calculations. If the lag time differs, the authors should redo their MFPT calculations with the updated lag time.

REPLY:

The Δt parameter used in the model by Pande et al. plays a different role as compared to the lag time used in the MSM. For the Pande et al. method, using different parameters Δt in Eq. 1 had only a marginal effect on the computed MFPTs, if any. (We tested values between 0.1 and 20ns). To avoid this confusion, we now refer to Δt as "time delay" and use the term "lag time" only in the context of the MSM. In addition, we now write:

"Using other time delays between 0.1ns and 20ns led to nearly identical estimates for the MFPTs."

In contrast, for the MSM, based on the implied timescale plot (new Figure S10C), we used a lag time of 20ns. For more details on the MSM, please see our long reply below.

REVIEWER:

The authors have a great opportunity here for generating a Markov state model (MSM) from adaptively sampled data obtained using an enhanced sampling technique. This could be a great example case study for MD practitioners, regardless of their interest in helicase.

REPLY:

Many thanks for raising these important points, which were indeed not sufficiently addressed in the previous version of the manuscript. Since these comments on the Markov state model (MSM), the free energy landscape, and convergence related, we provide a longer joint answer:

The key findings of this study is the continuous conformational cycle which (i) provides atomic details of molecular switches and (ii), based on a simple kinetic model, overall kinetics that exhibit good order-of-magnitude agreement with the kinetics of related helicases. However, we do not claim that we have obtained a fully converged free energy landscape, which would be extremely difficult for such complex transitions. For instance, considering that we obtained only a single transition of the sensor serine during the opening from 300 independent simulations, one cannot expect a converged free energy difference between the states before and after the sensor serine transition.

However, we also fully agree with the reviewer that the construction of the Markov State Model (MSM), including an (at least crude) estimation of the underlying free energy landscape, adds to our study. Therefore, following this and the other reviewers' suggestion, we have derived the MSMs for the opening and for closing process and computed the respective free energy landscape, as shown in the new Supplementary Figure S9. The MSM separates the meta-stable states reasonably well and identified the respective long-living conformations (sub-panels of Figure S9). Various validation measures look reasonable (VAMP-2 score, implied times scales, and Chapman-Kolmogorov test, new

Figures S10 and S11). Furthermore, the MFPTs of the closing transition are in reasonable agreement with the results from the simple linear kinetic model.

In contrast, the MSM overestimated the rate for the opening transition (from state S1 to state S5 in the new Figure S9), most likely owing to lack of sampling of the loop-to-helix transition of the sensor serine, which we observed only once in 300 simulations. We now carried out extensive trials for improving the opening rate predicted by the MSM: we tried different lag times, number of microstates and macrostates, different features (with or without RMSD values or domain rotation angles, and we added another set of 160 simulations of 100ns. However, although the MSM was quite sensitive with respect to these parameters (as seen by others, <https://doi.org/10.1063/5.0039144>), the estimated opening MFPT persisted. Hence, we conclude that by far longer simulations would be required to obtain a fully converged MSM. In line with these expectations, Pande and coworkers recently used ~1.5ms of simulation to obtain a MSM of a kinase (~10x more than we used here), which likely involves simpler transitions as compared to the transition studied here (<https://doi.org/10.1038/s41557-018-0077-9>).

Hence, as emphasized previously by others (e.g., <https://doi.org/10.1063/5.0039144>), constructing a converged MSM for such complex conformational transition remains challenging. For the present manuscript, as the best service to the reader, we decided to present both kinetic models in the new manuscript, (i) the simple linear model based on the approach by Pande et al. as presented in the previous version of the manuscript, and (ii) the MSM. While the linear kinetic model is simple, intuitive, and numerically robust, the MSM is based on a more elaborate theory yet possibly subject to some bias owing to limited sampling. These points are now discussed, together with a warning that the numerical details of the MSM must be interpreted with care. The kinetic models are now presented in a separate section "Kinetic models of opening and closing processes".

OTHER COMMENTS

3. The authors should explain the aggressiveness of their adaptive sampling regime, and how their selection strategy does introduce increased bias into their simulations. Specifically, they should point out that their adaptive sampling strategy employs Supervised MD (SuMD). However, they should show that the extent of their sampling and its incorporation into a master kinetic framework (like a Markov state model) could remediate the bias introduced by seed state selection.

a. Extent of reweighting achieved by the Markov state model can be determined by showing that there is a linear relationship between the raw and MSM counts within the data. Basically, the log raw probability that a frame belongs to a cluster number should be roughly the same as the same frame's MSM-weighted probability of belonging to that same cluster. This type of analysis, as well as general MSM construction, is very reliant on good state discretization (feature-based clustering). This type of analysis has been observed in the following studies:

i. <https://www.nature.com/articles/s41557-018-0077-9#Sec12>, Supplemental Figure 2b

ii. [https://www.jbc.org/article/S0021-9258\(22\)00204-6/fulltext](https://www.jbc.org/article/S0021-9258(22)00204-6/fulltext), Supplemental Figure 21B

iii. <https://www.biorxiv.org/content/10.1101/2022.10.12.511964v1>, Supplemental Figure 11

REPLY: We agree that, for validating a *converged* MSM, the analysis would be important. However, since we do not claim that the MSM and the free energy landscape are fully converged, we suggest that the presentations of implied timescales, the VAMP-2 score, and the Chapman-Kolmogorov test are appropriate to show that the free energy landscape is a good approximation.

4. Concatenating the “successful” trajectories from the adaptive sampling regime is a really creative way to capture a single trajectory from the aggregate dataset, as well as make the results more appealing to an experimental audience. However, Figures 3 and 5 are extremely busy and have layouts which do not feel very intuitive at first glance. Is there a way these panels can be reformatted to make them easier to read? Aside from this, the time series from Figure 3 and Figure 5 could benefit from additional labels at critical events. For example, there is discussion about hydrogen bond rearrangements for some of these panels but it is hard to compare the manuscript main text to these figures; it is easy to get lost when looking at all the panels.

REPLY: Thank you for this suggestion. Following the reviewer, we have added more labels to Figures 3 and 5.

5. The snapshots and protein figures in this manuscript have been rendered very nicely. If needed for Figure 3 and 5 improvement, it could be a nice addition to provide expanded snapshots of these individual events with reference to the entire protein structure. These snapshots could be added to the SI so that these events can be structurally contextualized with the rest of the protein.

REPLY: We agree. We included the Figures as suggested as Supplementary Figures S12 and S13. In addition, we have labeled the H-bond partners in the RNA movie and enhanced its overall quality.

6. The SI for this manuscript submission was botched somehow. All the SI figures are embedded into the pdf, but none of the figures were labeled with titles or captions. Instead, I had to open peripheral files in a text editor and compare it to the order of the figures. Additionally, this made it very difficult to properly interpret what was going on during the MD movie files. Hopefully this can be fixed for the resubmission.

REPLY: We apologize for these problems. This might be a result of the submission process, since the submission website did not allow a separate supplementary files. We will fix this issue for the new submission.

7. Would it be worthwhile to make mp4 movie files capturing the entire concatenated trajectories that were made from the adaptive sampling? This could add to the structural emphasis of this study.

REPLY: In fact, we submitted such a movie with already with the previous manuscript, hence it should (hopefully) have been accessible for the reviewer. However, we will double-check that the movie is properly submitted during the revision.

8. The authors used a very strong restraint force during their very short equilibration runs (1000 kJ/mol*nm² for a 100 ps run). Are there other studies which use such a short equilibration simulation length? Would it not have been worthwhile to perform an equilibration without positional restraints?

REPLY: Thank you for this suggestion. Since the system equilibrates rapidly during the simulated tempering during the initial adaptive sampling simulations, we have indeed not carried out an additional equilibration in this study. However, we will follow this suggestion in our next simulations.

9. There are a few typos in the manuscript, as well as inconsistent references to figures. For instance, sometimes figures are referenced as “Figure X”, whereas other times the same figure is referenced as

“figure X”. All figures and tables should be referenced with capital (uppercase) letters, including Supplemental Figures (or at least some consistency should be maintained).

REPLY: Thank you, we have corrected these inconsistencies and carried out another round of proof reading and spell checking.

Reviewer #2 (Remarks to the Author):

This manuscript reports an interesting study on the molecular dynamics simulations of a DEAH-box helicase, which is an exemplary molecular motor protein. By combining simulated tempering (ST) and adaptive sampling (AS), the authors characterized the millisecond-scale conformational transitions between the open and closed states of the motor domains (RecA1 & RecA2) of Prp43. Interestingly, they found that loop-to-helix (helix-to-loop) transition of the sensor loop is a rate-limiting process for the opening (closing) of the RecA1-RecA2 domain interface. The all-atom simulations reveal the motions of key motifs that fit well with established structural models. In general, the manuscript is well-written and interesting to the helicase research community. However, the following issues should be addressed before the manuscript can be accepted for publication in Commun. Biol.

1. The author should calculate the free energy profiles projected along the structural features, such as a 2D free energy profile obtained using RecA1-RecA2 and S387-I383 distances shown in Fig. 3O and 5G. As each round of adaptive sampling has many independent simulated tempering simulations, the weighted histogram analysis method (JCTC, 2007, 3, 26-41) can be applied to obtain the free energy profiles. This analysis is important to locate intermediate states and check the convergence of the AS protocol.

REPLY:

Thank you for this suggestion, which was (in a similar spirit) also raised by Reviewer 1. To estimate the underlying free energy landscape, we have constructed a Markov State Model (MSM, see new Supplementary figures 9-11). The free energy landscape reveals several metastable states. However, as discussed in our reply to Reviewer 1 above, please note that we do not claim that we obtained a fully converged free energy landscape. Please note our longer reply to Reviewer 1 above for a more detailed discussion.

2. The convergence of the AS protocol should be demonstrated. For example, Ref. 40 checked the AS convergence by calculating a key observable (binding free energy) over multiple iterations.

REPLY:

As discussed in the long response above to Reviewer 1 above, we do not claim that the free energy landscape sampled by the AS is fully converged. Specifically, for the rate-limiting loop-to-helix transition of the sensor serine, we observed only a single transition within 300 independent simulations, demonstrating that obtaining exhaustive sampling would be extremely difficult here. Instead, the key result of this study is the derivation of a continuous transition of the opening and closing pathways with kinetics that are, remarkably, in reasonable agreement with experimental data for other helicases. To make clearer that we do not claim having obtained exhaustive sampling, we now write in the Discussion:

"In addition, AS is suitable for estimating the kinetics of the overall pathway by collecting the rates between neighboring metastable states. For the successful sequence of transitions, using a simple linear kinetic model, we estimated the MFPTs of the opening and closing transitions in the order of 1ms or 0.5ms, respectively (assuming tenfold accelerated rates owing to the use of ST). These values are in reasonable agreement with experimental data for other helicases [50–52]. Complementary, we constructed a MSM for the conformational cycle. While the MSM suggested a closing rate in reasonable agreement with the linear kinetic model, the MSM overestimated the opening rate, which we ascribe to insufficient sampling of the rate-limiting loop-to-helix transition of the sensor serine. Hence, for complex transitions as studied here, for which constructing a converged MSM remains challenging, the linear kinetic model provides a numerically robust and useful alternative."

3. In the method section of mean first passage time (MFPT) estimation, it is unclear how the N=8 states were obtained (such as coordinates used for clustering). What are the features of each state (e.g., RecA1-RecA2 distance, sensor loop, hook-loop, arginine finger)? The MFPT calculations assumed the non-neighboring intermediate states have no transitions. This assumption can be verified after assigning each frame of a trajectory to a specific state.

REPLY:

Thank you for raising this point. First, we apologize for mentioning N=8 states in the previous manuscript - in fact, we modeled the opening and closing pathways by N=7 states and six transition. This has now been corrected (see also Supplementary Table S1). Molecular views on these transitions are shown in Supplementary Figure S7.

Regarding the selection of the states, we did not use any automated procedure but instead selected the states by

- * extensive visual inspection of the trajectories

- * extensive inspection of "heat maps" as shown in Figure S1, which originally involved far more (putative) features than features finally used in our study.

For the interest of the reviewer, we believe that the extensive human supervision of the simulations was critical for the successful simulation of the overall transition. From these extensive visual inspections, we did not observe transitions between non-neighboring states, suggesting that they are extremely rare. Furthermore, since the individual transitions were highly interdependent, having such transitions between non-neighboring states seem unlikely anyway.

4. For the opening simulation, the target values of the structural features were taken from the open structure of Prp22, a homolog to Prp43. How similar are Prp22 and Prp43 in terms of structure and sequence? Sequence alignment should be performed to check the similarity between Prp22 and Prp43. Are the residues used for the structural features conserved in the two helicases?

REPLY:

Thank you raising this point. We added the following to the Methods:

"The helicase Prp22 is homologous to Prp43 studied here, as demonstrated by sequence identity and similarity of 47% and 63%, respectively, according to a FASTA sequence alignment \

cite{pearson_rapid_1990}. Furthermore, residues of the selected features are conserved among Prp22 and Prp43, except for G349 in Prp43 that is replaced with serine in Prp22. However, since G349 interacts with the RNA mostly via the protein backbone, this replacement has only a minor effect on the characterization of the opening transition. Hence, target values for the selected features were taken from the open Prp22 structure (PDB ID 6I3P\cite{Hamann2019})."

5. The polarity of Prp43 should be discussed in the manuscript. It is known that DEAH-box proteins translocate in the 3'→5' direction. Are the simulations able to illustrate the molecular basis for this polarity?

REPLY: This is an interesting point, which we have indeed discussed intensely during this project. However, because it is difficult to make a quantitative statement on the mechanism of the directed transport (the polarity), we have not discussed this point in the previous manuscript.

For the interest of the reviewer, it is tempting to speculate that the transition of the serine loop, followed by the formation of an additional H-bond with the RNA, plays a key role. During the opening process, all H-bonds between RecA2 and RNA shift one nucleotide upstream, which may require a slightly weaker RecA2-RNA binding interface. Upon the serine flip, an additional RecA2-RNA H-bond forms, thereby strengthening the RecA2-RNA interaction. After insertion of ATP, this additional RecA2-RNA interaction may be required to (at least temporarily) maintain the tight RecA2-RNA binding, as required for pushing the RNA along the RecA1 domain.

However, quantifying the RecA2-RNA interaction merely from the number of H-bonds to support such mechanism, seem too speculative in our opinion. In fact, we carried out some attempts to quantify the RecA2-RNA (by potential energies), however, the results were hard to interpret. Therefore, we prefer not to speculate and merely report that the AS simulation revealed the directed transport, in agreement to experimental data.

Minor issues:

1. The RMSD values from the target states should be measured to monitor how far the system is from the target at the end of each trajectory.

REPLY: We fully agree. We have added a new Supplementary Figure S8 with the RMSD during the opening and closing processes.

2. In the last paragraph of Discussion, a computational study (JACS, 2015, 137, 3031-3040) of an RNA helicase should be mentioned, as it employed pathway sampling methods (the string method and milestoning) to study the motor translocation. The MFPT calculation by milestoning, which is rather rigorous, revealed a 0.11-millisecond conformational transition triggered by ATP hydrolysis product release.

REPLY: Thank you, we have added the reference into the Discussion section.

3. The authors might want to be more careful when comparing the calculated MFPTs to the experimentally measured translocation speed. It has been shown for other helicase systems that phosphate or ADP release is the rate limiting process for the functional cycle. The powerstroke (conformational change) can be orders of magnitude faster than the hydrolysis product release process.

REPLY: Thank you for pointing this out. Since we fully agree, we now write more carefully: "This implies a maximum translocation speed of ~ 600 base pairs per second (bp/s). Owing to contributions from ATP binding, hydrolysis, and release, which may even be rate-limiting for the overall cycle, the translocation speed of Prp43 is likely lower than the maximum value of ~ 600 bp/s estimated from our simulations. Notably, assuming some reduction of the translocation speed from ATP binding, hydrolysis and release, our value is in reasonable agreement with translocation speeds of 100 to 300 bp/s observed for other helicases."

4. The last paragraph of Methods talked about the increased computational efficiency. How many days did it take to complete the AS protocol for opening and closing transitions (with trivial parallelization and multiple GPUs)?

REPLY: The number of simulation days strongly depends on the hardware and the number of accessible GPUs. To obtain a very approximate value, by the start of the study, we used 40 six-core nodes with on Nvidia RTX 1070Ti GPU each. With a performance of 40 nanoseconds a day, the simulations presented here would require 60 days under full load. Hence, the simulations presented here are costly yet increasingly feasibly on more modern hardware.

5. It would be great to make a movie to show the details of the helix-to-loop/loop-to-helix transitions of the sensor loop (zoom-in view).

REPLY: We agree. We have updated the RNA movie to make the serine loop more visible.

6. - Table 2 is missing.

- In the first paragraph of Page 23, "56.500 ns" and "40.000 ns" should be "56.5 μ s" and "40.0 μ s", respectively.

- Need captions for the supplement figures.

REPLY: We apologize for this mistake. We changed the units accordingly.

The SI Figures seemed to be corrupted due a Latex error in our submission process. The original file were correct. We will double-check for such problems during the submission of the revised manuscript.

Reviewer #3 (Remarks to the Author):

Summary: Prp43 is a DEAH box helicase involved in the translocation of prokaryotic single stranded RNA (ssRNA). Although the mechanism of translocation in Prp43 is somewhat understood through their X-ray crystal structures, the actual sequence of intermediate events is still unknown. Through the present work, the authors utilized the concerted power of Adaptive Sampling (AS) and Simulated Tempering (ST) molecular dynamics (MD) simulations to capture atomistic details of the conformational cycle of Prp43. While ST helps in lowering the energetic barriers using higher temperatures, AS help in conformational sampling of rate-limiting steps as well as parallelize such sampling, thereby gaining computational efficiency over conventional MD. The authors used crystallographic data as start and end points, as well as crystallographic features to guide the course of the simulations. The work demonstrates that combination of the two enhanced sampling methods could help interpret subtle but kinetically important events, saving folds of computational times over conventional MD. Additionally, the detailed atomistic study of Prp43 helicases suggested a modification to the conventional inchworm motion of the RecA domain over RNA. The authors

suggest that while the center of mass of RecA follows an inchworm like motion over RNA, the domains rather follow a caterpillar like crawling movement. Thus, the authors demonstrated that large scale conformational changes are guided by small scale atomistic events, which could be captured using a combination of the AS and ST methods.

Significance:

To date, multiple authors have worked on using all-atom simulations to decipher the dynamics of motor proteins¹⁻³. Additionally, the crystal structures of the RNA bound DEAH box helicases such as Prp43 were only recently resolved, and the conformational cycle was outlined⁴. Despite this, the intermediate steps in the conformational cycle of such helicases is still unknown. Given the size of the helicase-RNA-ATP/ADP complexes, simulation of conformational changes at an atomic scale incurs unrealistic computational cost. To address this, the authors suggest utilizing a combination of two enhanced sampling methods such as AS and ST, to overcome the energy barriers in the conformational cycle. As communicated through the article, the significance of the work is threefold:

- a. Utilization of two enhanced sampling methods such as Adaptive Sampling (AS) and Simulated Tempering (ST) to enhance computational efficiency in sampling rare events in the complex conformational cycle of a large motor protein.
- b. Importance of all-atom simulations in deciphering the key steps in conformational cycle of large biomolecular complexes, as illustrated by the role of the key biomolecular residues.
- c. Essential residues labelled as “molecular switches” are identified through the present work. Although the roles of such switches are well identified through X-ray crystal structures, the actual sequence of events is not understood in detail. Such details could be used in the further study of DEAH box helicases.

The authors have presented a clear and detailed description of the work, including the application of the two methods and explanation of the corresponding results. The justification of choosing the parametric features for the simulations has been discussed and compared with the experimental start and endpoints for validation. Overall, the work presented by the authors is entirely novel and could be applied to study conformational dynamics of large biomolecular systems, and also help understand molecular mechanisms of motor proteins.

REPLY:

Many thanks for the detailed and positive assessment of our study.

REVIEWER:

Comments:

The reviewer has the following comments related to the article:

1.Modification of the title: The title “Continuous all-atom conformational cycle of a motor enzyme” suggests that the main purpose of the article is achieving an all-atom simulation of a motor enzyme. This is although somewhat true, does not fully bring out the potential of the work communicated through the article. The reviewer suggests that the title of the article be modified to capture the interest of readers who would want to apply such methods on their systems and to address those who would be interested in studying Prp43.

REPLY: Thank you for pointing this out. Following the reviewer, we changed the title to provide a more specific overview on our study: "Continuous millisecond conformational cycle of a DEAH box helicase reveals control of domain motions by atomic-scale transitions"

2. Labelling residues in videos for clarity to the discussions:

The reviewer thinks that major residues should be labelled in the videos for clarity to the readers. For example, the authors have clearly described the role of K403 H-bond interactions with U4 and U3 as the molecular switch, through discussions in text and images in Figure 3 (A-C). However, labels of such residues in the video are missing. Labelling these residues/side chains in the videos will allow the readers to coherently relate the text and figure 3 to video 1.

REPLY:

We fully agree. We have now improved Movie S1 with labels and with a closer view on the serine loop. In addition, we added more labels to Figures 3 and 5 to relate the figures more clearly to the text.

REVIEWER:

3. Role of water molecules:

a. The role of a water molecule in engaging the arginine finger (R435) has been mentioned by Tauchert et al. (2017), who resolved the crystal structure of Prp43-ADP-BeF3-U7-RNA complex 4. The reviewer is curious if the authors found any such water mediated mechanism related to the anchoring of the arginine finger.

REPLY:

We added a new Supplementary Figure S14 to show the water network at the end of the closing process. At the end of the closing process, the water network is similar to the network reported by Tauchert et al., with three water molecules in contact with the Mg²⁺ (thick sticks in Fig. S14). However we did not observe the water molecule reported by Tauchert et al. in nearly perfect position for a Sn2 attack. This is now described in the figure legend of Fig. S14.

REVIEWER:

b. Additionally, although the authors mentioned the role of a water molecule in the stabilization of the closed state, by engaging S387 of RecA1, the respective figure 5E does not depict the water molecule. “where S387 interacts with a water molecule of the ATP–Mg²⁺–water complex to stabilize the closed state (Fig. 5E).”

REPLY:

The complex is now shown in Figure S14. Three water molecules coordinating the Mg²⁺ ion are highlighted as thick sticks.

4. Role of buffer solution:

The conformational dynamics of nucleic acids are well known to be influenced by the presence of buffer solutions such as KCl and/or MgCl₂. The importance of metal cations and buffer solutions have been found to be important in the activity of helicases via multiple studies 5-7.

Moreover, the experimental conditions included the addition of 150 mM KCl and 3 mM MgCl₂ at a pH of 7.4 4.

Despite this, the method section for the preparation of the system only includes addition of neutralizing ions, but not buffer:

a. The reviewer is curious if this is only a typo, and the authors did add a buffer during solvation. If otherwise, what was the rationale behind omitting the buffer solution(s)?

REPLY:

In fact, during an early state of this project, we discussed extensively the use of an ionic buffer (such as 150mM NaCl) versus the use of only counter ions. Where as real buffer is certainly slightly more realistic, we were concerned about more frequent long-living binding events of ions to the large number of charged moieties in the simulations, which could (putatively) lead to increased sampling challenges. Since sampling was the by far largest challenge in this study, we opted for the slightly simplified model with purely counter ions.

We did not add magnesium ions (except for the Mg²⁺ at the ATP site) since (non-polarizable) models of divalent ions are still somewhat controversial and may lead to unrealistic binding to the RNA backbone.

REVIEWER:

b. Is it possible that the observations with the electrostatic switches might have been different if buffer was present? Could the “crawling” instead of a typical “inchworm” motion be related to the absence of a buffer?

REPLY:

Although we did not investigate the effect of 150mM salt explicitly, it is unlikely that salt (on top of counter ions) would change the qualitative results of this study. 150mM salt (leading to a Debye length of ~1nm) would enhance the electrostatic shielding only between charged groups that are mostly separated by solvent (with a thickness of approximately 1nm or more). Instead, the key molecular switches involve short-range interactions at the RecA1-RecA2 or at the protein-RNA interfaces, suggesting that electrostatic shielding has a smaller effect. Furthermore, it is important to note that even counter ions alone enable some electrostatic shielding as well.

REVIEWER:

5. Flowcharts for the methods of AS and ST:

Although the authors have described the methods of AS and ST very clearly, a flowchart of the various steps of the two methods will allow the reader to comprehend better. These flowcharts could be added as supplementary information.

REPLY:

Thank you for this suggestion. We added flow charts of AS and ST as new Supplementary Figs. S15 and S16.

References mentioned in comments:

1. Hwang W, Lang MJ, Karplus M. Kinesin motility is driven by subdomain dynamics. *Elife*. 2017;6.
2. Shi, X.X., Wang, P.Y., Chen, H. and Xie, P., 2021. Studies of conformational changes of tubulin induced by interaction with kinesin using atomistic molecular dynamics simulations. *International journal of molecular sciences*, 22(13), p.6709.

3. Flechsig, H., Popp, D. and Mikhailov, A.S., 2011. In silico investigation of conformational motions in superfamily 2 helicase proteins. *PLoS One*, 6(7), p.e21809.
4. Tauchert, M.J., Fourmann, J.B., Lührmann, R. and Ficner, R., 2017. Structural insights into the mechanism of the DEAH-box RNA helicase Prp43. *Elife*, 6.
5. Brennan, C.A., Steinmetz, E.J., Spear, P. and Platt, T., 1990. Specificity and efficiency of rho-factor helicase activity depends on magnesium concentration and energy coupling to NTP hydrolysis. *Journal of Biological Chemistry*, 265(10), pp.5440-5447.
6. Cao, X., Li, Y., Jin, X., Li, Y., Guo, F. and Jin, T., 2016. Molecular mechanism of divalent-metal-induced activation of NS3 helicase and insights into Zika virus inhibitor design. *Nucleic acids research*, p.gkw941.
7. Frick, D.N., Banik, S. and Rypma, R.S., 2007. Role of divalent metal cations in ATP hydrolysis catalyzed by the hepatitis C virus NS3 helicase: magnesium provides a bridge for ATP to fuel unwinding. *Journal of molecular biology*, 365(4), pp.1017-1032.

REVIEWERS' COMMENTS:

Reviewer #1 (Remarks to the Author):

Authors have successfully addressed all the comments raised in the initial review.

Reviewer #2 (Remarks to the Author):

The authors have addressed all of my questions. I think the revised manuscript is ready for publication.